# Sequential role of RAD51 paralog complexes in replication fork remodeling and restart

Matteo Berti[1,4], Federico Teloni [2,4], Sofija Mijic [1], Sebastian Ursich[1], Jevgenij Fuchs [1], Maria Dilia Palumbieri[1], Jana Krietsch[1], Jonas A. Schmid[1], Edwige B. Garcin [3], Stéphanie Gon [3], Mauro Modesti [3], Matthias Altmeyer [2✉] & Massimo Lopes [1✉]

Homologous recombination (HR) factors were recently implicated in DNA replication fork remodeling and protection. While maintaining genome stability, HR-mediated fork remodeling promotes cancer chemoresistance, by as-yet elusive mechanisms. Five HR cofactors – the RAD51 paralogs RAD51B, RAD51C, RAD51D, XRCC2 and XRCC3 – recently emerged as crucial tumor suppressors. Albeit extensively characterized in DNA repair, their role in replication has not been addressed systematically. Here, we identify all RAD51 paralogs while screening for modulators of RAD51 recombinase upon replication stress. Single-molecule analysis of fork progression and architecture in isogenic cellular systems shows that the BCDX2 subcomplex restrains fork progression upon stress, promoting fork reversal. Accordingly, BCDX2 primes unscheduled degradation of reversed forks in BRCA2-defective cells, boosting genomic instability. Conversely, the CX3 subcomplex is dispensable for fork reversal, but mediates efficient restart of reversed forks. We propose that RAD51 paralogs sequentially orchestrate clinically relevant transactions at replication forks, cooperatively promoting fork remodeling and restart.

[1] Institute of Molecular Cancer Research, University of Zurich, Winterthurerstrasse 190, 8057 Zurich, Switzerland. [2] Department of Molecular Mechanisms of Disease, University of Zurich, Winterthurerstrasse 190, 8057 Zurich, Switzerland. [3] Cancer Research Center of Marseille; CNRS; Inserm; Institut Paoli-Calmettes; Aix-Marseille Université, 27 Bd Leï Roure, CS 30059, 13273 Marseille, France. [4] These authors contributed equally: Matteo Berti, Federico Teloni. ✉email: matthias.altmeyer@dmmd.uzh.ch; lopes@imcr.uzh.ch

Protecting the integrity of replicating chromosomes is crucial to maintain genome stability and to avoid cellular transformation. Mutations impairing the replication stress response predispose to cancer[1]. However, the same mutations often impair the cellular responses to genotoxic treatments, offering important windows of opportunity for effective cancer chemotherapy by DNA replication interference[2].

Replication fork reversal—i.e., the active conversion of replication forks into four-way junctions—has recently emerged as a global and genetically controlled response to various forms of endogenous and exogenous replication stress[3–5]. These transient replication intermediates can be efficiently restarted[6,7] and their function was proposed to slow down replication forks under unfavorable conditions, thereby limiting ssDNA accumulation and fork breakage[8–10]. Replication fork reversal requires specialized enzymes, such as the DNA translocases SMARCAL1, ZRANB3, and HLTF[11–14], and the central recombinase RAD51[5]. Genetic modifications affecting fork reversal typically result in unrestrained fork progression upon genotoxic stress, highlighting the relevance of fork remodeling for active fork slowing[5,12,15]. RAD51 function in regulating fork progression and remodeling extended the role of homologous recombination (HR) factors from classical double-stranded break (DSB) repair to the replication stress response[16]. Accordingly, other HR factors—like BRCA1 and BRCA2—protect stalled forks from excessive nucleolytic degradation by promoting efficient RAD51 loading[9,17,18]. This function in replication stress is genetically uncoupled from their canonical role in DSB repair[14,17,19], is associated with the exquisite chemosensitivity of BRCA-deficient cells[20] and has been recently extended to additional HR factors[21]. Importantly, this deregulated degradation of nascent DNA is also genetically dependent on fork remodeling, as regressed arms are mandatory entry points for DNA degrading enzymes[13,14,22,23]. Moreover, the HR protein RAD52 was recently shown to modulate fork stability and restart, by limiting SMARCAL1-dependent fork reversal or regulating reversed fork processing by MRE11[14,24]. Collectively, these recent observations have implicated the RAD51 recombinase and other HR factors both in the formation and in the stability of reversed replication forks[5,14]. The latter function is BRCA2-dependent and reflects protection of the DNA end at the fourth regressed arm. However, the role of RAD51 in reversed fork formation is still enigmatic, as fork reversal does not require BRCA2, nor stable RAD51 nucleofilament formation[14], thereby challenging canonical models of RAD51 function[16]. Molecular understanding of these clinically relevant transactions on replicating chromosomes requires further mechanistic investigation, particularly on the role of alternative HR mediators and on their functional interaction with RAD51.

Besides RAD51 and its meiotic counterpart DMC1, five additional mammalian paralogs of bacterial RecA were discovered two decades ago, either by DNA sequence alignments (RAD51B, RAD51C, and RAD51D) or by functional complementation of radiation sensitive Chinese hamster ovary (CHO) mutant cells (XRCC2 and XRCC3)[25]. These proteins display limited sequence homology to each other and to RAD51, and are generally reported as classical RAD51 paralogs. The Shu complex, which regulates HR in mitosis and meiosis, was recently shown to include a sixth RAD51 paralog, i.e., SWSAP1[25–27]. Classical RAD51 paralogs were proposed to form two biochemically and functionally distinct subcomplexes, i.e. the RAD51B-RAD51C-RAD51D-XRCC2 complex (BCDX2) and the RAD51C-XRCC3 complex (CX3), showing common, but also distinct biochemical properties[28–32]. Ablation of these proteins in mice resulted in embryonic lethality and functional studies on these factors were thus largely based on gene inactivation in p53-deficient CHO cells

and chicken DT40 B-lymphocytes[33–39]. Overall, these studies uncovered important roles for these proteins in genome stability and DSB response, via modulation of RAD51. Recent biochemical data on the dimeric RAD51 paralog complexes in *S. cerevisiae* and *C. elegans*—Rad55/Rad57 and RFS-1/RIP-1, respectively—supported this concept, by showing that RAD51 paralogs protect RAD51 nucleofilaments from antirecombinase activities[40] and remodel these filaments to promote strand exchange reactions[41,42]. However, whether similar biochemical properties can be associated with the different subcomplexes of human RAD51 paralogs is still unknown.

siRNA-mediated depletion of the human proteins suggested different roles for the two RAD51 paralog subcomplexes at early (BCDX2) versus late (CX3) stages of HR-mediated DSB repair[43–45]. Very recently, a panel of RAD51 paralog gene ablations was produced by CRISPR-Cas9 in cancerous and non-transformed human cell lines[46]. While *RAD51B* inactivation lead to comparatively mild phenotypes, genetic ablation of any of the other paralogs resulted in drastic impairment of HR-mediated DSB repair proficiency in all tested cell lines[46].

Mutations in several RAD51 paralog genes have been associated with increased susceptibility to breast and ovarian cancer[47–51]. Moreover, hypomorphic mutations in *RAD51C* and *XRCC2* were linked to Fanconi anemia, a rare human disorder linked to defective replication-coupled repair of specific DNA lesions in the bone marrow[52,53]. These recent findings revived the interest in these accessory HR factors, promoting new mechanistic studies to unravel their precise role in DNA replication and genome stability. Interestingly, all human RAD51 paralogs were shown to associate with nascent DNA[54], but the mechanistic role (s) of these factors in replication were not investigated systematically. CHO or DT40 cell lines carrying different mutations in individual genes—i.e., *RAD51C*, *XRCC2*, and *XRCC3*—displayed specific defects in replication fork progression and stability[54–56]. Furthermore, based on shRNA-mediated downregulation, human XRCC2 was recently proposed to modulate fork progression also in human cells, as a specific response to limited nucleotide availability[57]. Overall, our mechanistic understanding of the role of human RAD51 paralogs at replication forks is still incomplete and requires a systematic analysis of these factors in isogenic backgrounds.

Here, all classical human RAD51 paralogs are identified in a high-content microscopy screen for RAD51 modulation upon mild replication stress. Combining these cytological data with single-molecule investigations on replication intermediates in multiple cellular systems, we find that the BCDX2 complex assists RAD51 in reversed fork formation, limiting fork speed upon DNA damage and mediating stalled fork degradation in BRCA2-defective cells. Conversely, the CX3 complex—albeit dispensable for fork reversal and fork protection—is found to mediate efficient restart of previously reversed forks.

## Results

### A screen for RAD51 foci in CPT-treated cells identifies RAD51 paralogs.
RAD51 is readily detected on chromatin as nuclear foci in human osteosarcoma U2OS cells even during unperturbed replication. To identify key modulators of the RAD51 recombinase at challenged replication forks, we subjected U2OS cells to a short (45 min) treatment with low dose of the Topoisomerase I inhibitor camptothecin (CPT, 50 nM), which was previously shown to induce marked fork slowing and frequent fork reversal[58]. As in these conditions induction of DSBs is undetectable[58], RAD51 foci likely mark HR-mediated replication fork remodeling events occurring at stalled, yet-unbroken forks (Fig. 1a). We then designed a targeted small interfering RNA (siRNA) library, using

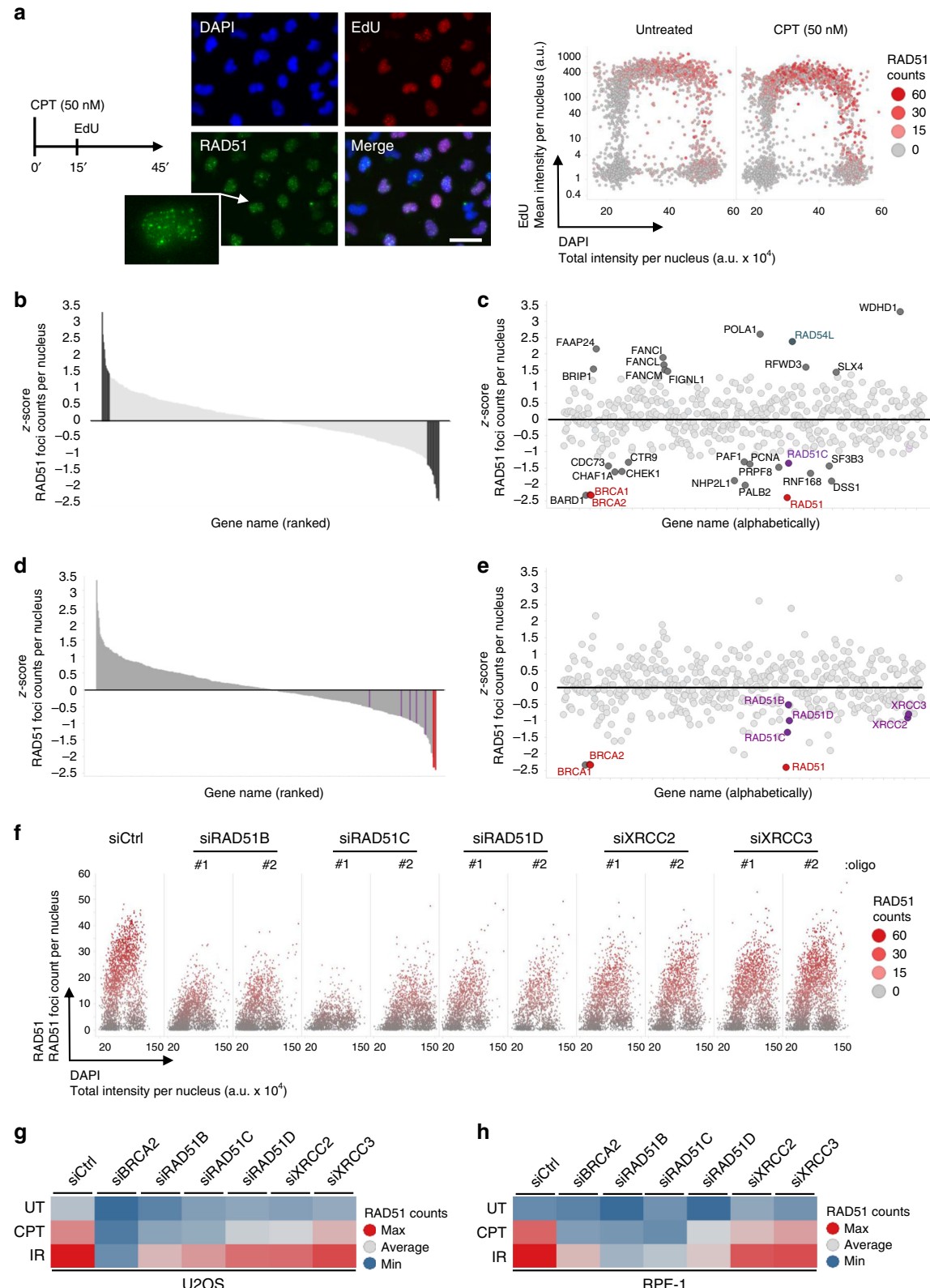

three individual siRNAs against each target gene and covering approximately 450 factors that were found to be enriched at replication forks by immunoprecipitation of newly synthesized DNA and mass spectrometric analysis of the associated proteins[59–61]. CPT-treated cells were labeled with EdU and analysed by high-content microscopy for quantitative image-based cytometry (QIBC)[62,63], which allows simultaneous assessment of cell-

cycle distribution, replication competence, as well as number and intensity of RAD51 foci. We selected an early time point after siRNA transfection (48 h), in order to limit cell-cycle effects due to prolonged depletion of essential factors. In these hypomorphic conditions, only depletion of key replication factors (e.g., PCNA) led to replication impairment, while most other siRNAs did not impact replication competence and cell proliferation, including

**Fig. 1 QIBC-based screen identifies RAD51 paralogs as important modulators of RAD51 foci formation upon mild replication stress. a** QIBC-based screen layout. Left: screen experimental condition. U2OS cell were treated with 50 nM CPT for 45 min and replicating cells were concomitantly labeled with EdU for 30 min. Center: representative immunofluorescence image (scale-bar, 50 μm). Right: representative scatter plot showing cell-cycle distribution (based on DNA content and EdU incorporation) of RAD51 foci in untreated and CPT-treated U2OS cells. The experiment shown was performed twice yielding similar results. **b** Screen gene ranking based on the RAD51 foci z-score. **c** z-score plot of positive (z-score < −1.5) and negative (z-score > 1.5) RAD51 modulators (marked in gray). Color code as in **d**, **e**. **d**, **e** z-score for the five classical RAD51 paralogs (marked in purple) and positive controls (marked in red). **f** Validation of the RAD51 paralogs influence on RAD51 foci formation upon 50 nM CPT. **g**, **h** Differential effect of RAD51 paralogs on RAD51 foci formation in untreated (UT), CPT- (50 nM for 45 min) and IR- (5 Gy, 2 h recovery) treated U2OS and RPE-1 cells. Source data for a-f are provided in the Source Data file.

those targeting essential HR factors like BRCA1 (Supplementary Fig. 1a). Reassuringly, when ranked according to their deviation from the mean (i.e., z-score), known positive (BRCA1, BRCA2, BARD1, PALB2) and negative regulators (RAD54L) of RAD51 scored among the top hits at either end of the distribution (Fig. 1b, c, Supplementary Data 1 and 2), with all three siRNAs showing consistent results (Supplementary Data 2). However, presumably because of the hypomorphic conditions chosen for the screen, siRNAs targeting genes of other known anti-recombinase activities (i.e., *RADX/CXorf57*, *FBH1/FBXO18*, *RECQ5*, and *PARI/C12orf48*) did not show strong effects on RAD51 accumulation (Supplementary Fig. 1b–c; Supplementary Data 1 and 2). Furthermore, although RAD51 has been previously involved in different steps of fork remodeling and protection, siRNAs targeting genes for key fork remodeling factors (i.e., *SMARCAL1*, *ZRANB3*, *HLTF*, and *RECQ1/RECQL*) did not significantly affect RAD51 foci number and intensities in our screening conditions (Supplementary Fig. 1b–c). Besides the possibility of incomplete downregulation, this result may also reflect intrinsic ambiguities associated with RAD51 foci measurements as sole readout. Impairing fork remodeling activities may not only negatively affect RAD51 binding to forks, but can also induce fork breakage even at these minimal CPT doses[5,58], thereby enhancing RAD51 accumulation in foci.

Despite these potential caveats and its inherently non-comprehensive nature, our screen identified several new or uncharacterized modulators of RAD51 chromatin binding upon replication stress, showing strong positive (>1.5) or negative (<1.5) z-scores (Fig. 1c, Supplementary Data 1). Here, we focused our attention on the five classical RAD51 paralogs, which all displayed a negative z-score in the experimental conditions of our screen (Fig. 1d, e). SWSAP1—encoding another noncanonical RAD51 paralog[25–27]—was not among the targeted genes in our siRNA library, but downregulation of its interaction partner *SWS1* led to defects in RAD51 foci comparable to those observed for the other RAD51 paralogs (Supplementary Fig. 1b–c, Supplementary Data 1), in agreement with the recently reported involvement of the Shu complex in the replication stress response[64].

In light of the established clinical relevance of the classical RAD51 paralogs and their elusive role in the replication stress response, we decided to focus on and explore this set of genes. We validated the screen results in U2OS cells using two individual siRNAs, testing also—besides low CPT treatments—untreated conditions and ionizing radiation. In these analyses, short-term (48 h) depletion of all classical RAD51 paralogs consistently led to reduced RAD51 foci counts in replicating cells, while not significantly affecting cell cycling and replication competence (Fig. 1f, g and Supplementary Fig. 1d). In agreement with previous reports[45], we noticed that *XRCC3* downregulation showed milder effects on RAD51 foci, compared to inactivation of the other RAD51 paralogs. This difference was also observed when performing validation experiments in untransformed human retinal epithelial cells (RPE-1, Fig. 1h) and is particularly

evident in S phase cells upon mild CPT treatment (Supplementary Fig. 1e). As XRCC3 and RAD51C form a specific protein complex (CX3), while all other factors assemble in a second multimeric RAD51 paralog complex (BCDX2), these results pointed to separate functional roles of the two complexes in the replication stress response, as suggested for DSB repair[45].

**BCDX2 promotes fork slowing and remodeling upon mild genotoxic stress.** To investigate the functional role of BCDX2 and CX3 in the replication stress response, we first monitored the stability of the two complexes upon single-component down-regulation in U2OS cells by two independent siRNAs (Fig. 2a, b). In keeping with previous reports[45], *RAD51C* downregulation affected protein levels of all components of both complexes, thus potentially abolishing both BCDX2 and CX3 activities. In contrast, *RAD51D* downregulation did leave CX3 levels unaffected, thus allowing the assessment of the specific functional role of BCDX2. Conversely, *XRCC3* downregulation had only minor effects on RAD51C levels and should thus specifically affect CX3 function (Fig. 2a, b). We therefore focused our functional analysis on these three protein depletions, to explore specific roles of the two complexes upon mild replication interference (Fig. 2a). We first used an established nascent DNA labeling protocol to monitor by DNA fiber spreading active replication fork slowing upon mild treatment with CPT[58]. *RAD51C* and *RAD51D* downregulation by two independent siRNA sequences drastically impaired active fork slowing, leading to unrestrained fork progression in the presence of CPT, both as measured by fiber track length and as ratios between the differentially labeled nascent DNA (Fig. 2c, and Supplementary Fig. 2a–c). Conversely, *XRCC3* downregulation did not significantly affect CPT-induced fork slowing (Fig. 2c, and Supplementary Fig. 2a–c). We then performed similar experiments in U2OS cells where the same three genes had been knocked-out (KO) by CRISPR-Cas9[46]. Compared to RAD51 paralog downregulation by siRNA (Supplementary Fig. 1a), all KO cells—including *XRCC3-KO*—displayed a more drastic reduction of endogenous and drug-induced RAD51 foci (Supplementary Fig. 2d), likely reflecting full and prolonged inactivation of these factors. Nonetheless, all tested KO cells led to very similar observations when compared to downregulated cells in terms of protein level interdependency and replication fork progression phenotypes, with *RAD51C-KO* and *RAD51D-KO* cells, but not *XRCC3-KO* cells, displaying unrestrained fork progression in CPT (Fig. 2d, e). Importantly, unrestrained fork progression in *RAD51C-KO* and *RAD51D-KO* cells was readily complemented by stable expression of a FLAG-tagged version of the missing protein (Fig. 2d, e). Finally, we confirmed very similar effects on CPT-induced fork slowing by siRNA-mediated down-regulation of *RAD51C*, *RAD51D*, and *XRCC3* in the non-transformed RPE-1 human cell line (Supplementary Fig. 2e). Overall, these data strongly suggest that BCDX2, but not CX3, mediates active fork slowing upon mild genotoxic stress.

Active fork slowing is linked to replication fork reversal (Fig. 2f) and depends on the RAD51 recombinase and multiple

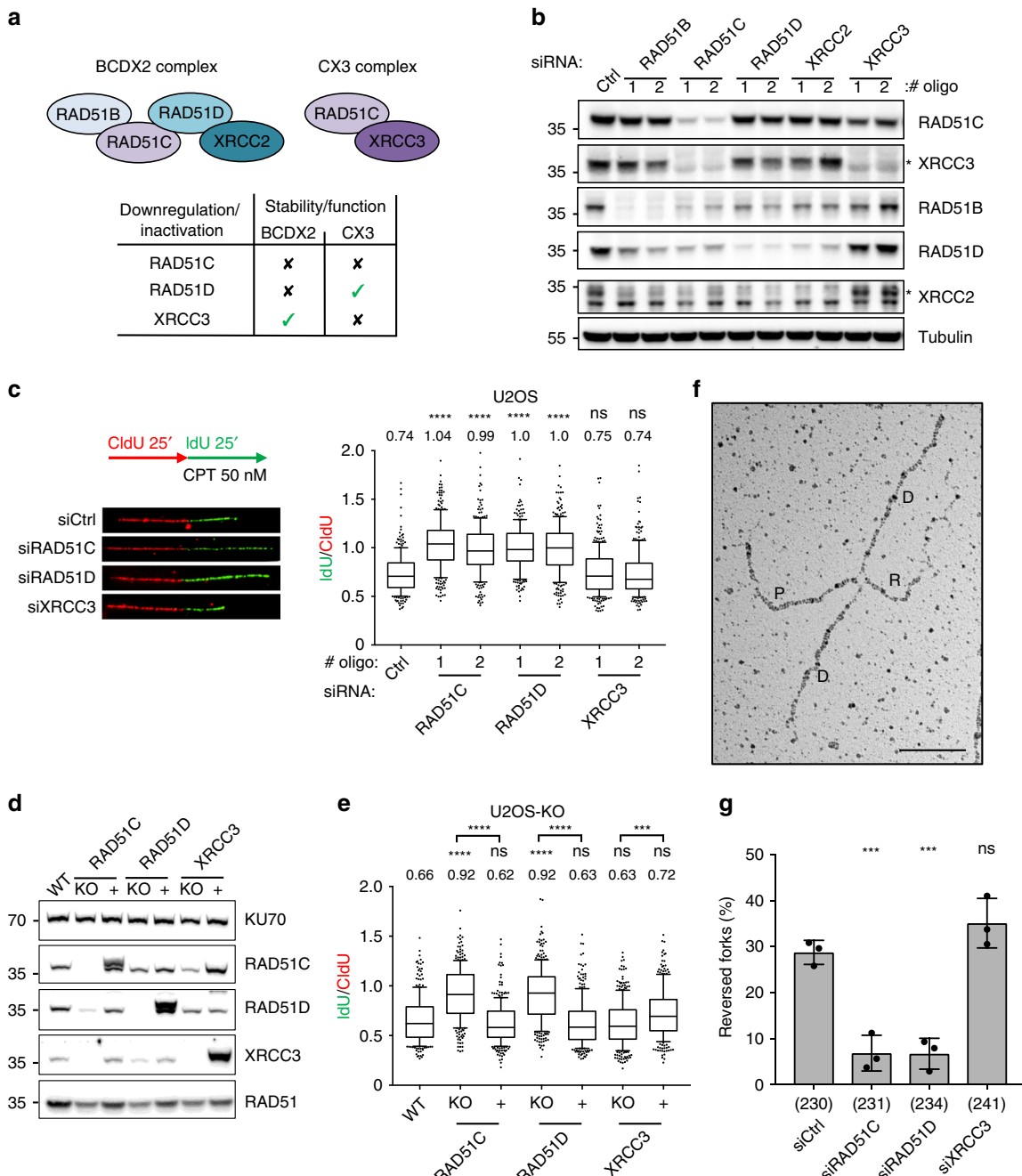

**Fig. 2 BCDX2 paralog complex, but not CX3, promotes replication fork slowing and reversal upon mild CPT treatment. a** Rationale for focusing on *RAD51C*, *RAD51D*, and *XRCC3* downregulation, to perform functional studies on BCDX2 and CX3 complexes during replication stress. Top: schematic of the two main RAD51 paralog complexes. Bottom: differential effects of *RAD51C*, *RAD51D*, and *XRCC3* inactivation on paralog complex stability/function. **b** Western blot analysis of RAD51 paralogs upon 48 h downregulation with two different siRNAs. Ctrl, control siRNA; Tubulin, loading control. asterisk, specific band. In **b** and **d** multiple gels/blots were processed in parallel, ensuring equal and comparable loading across gels. The experiment was performed twice yielding similar results. **c** DNA fiber analysis of U2OS cells 48 h after transfection with a control siRNA (siCtrl) or with siRNAs targeting *RAD51C*, *RAD51D*, and *XRCC3*. Top-left: schematic of the CldU/IdU pulse-labeling protocol used to evaluate fork progression upon 50 nM CPT treatment. Bottom-left: representative DNA fibers from each genetic condition. Left: IdU/CldU ratio is plotted as a readout of fork progression. In **c** and **e**, the numbers indicate the mean value; a minimum of 150 forks was scored in two independent experiments yielding similar results. Bounds of box are 25–75th percentile, center shows the median, whiskers indicate the 10–90 percentiles, data points outside this range are drawn as individual dots. Statistical analysis: Kruskal–Wallis test; ns not significant; ****$p$-value < 0.0001, ***$p$-value = 0.0003. **d** Western blot analysis of RAD51 paralogs in Knock-Out (KO) U2OS cells and in cells reconstituted of the respective protein (+). KU70, loading control. The experiment was performed twice yielding similar results. **e** DNA fiber analysis of KO and reconstituted (+) U2OS cells labeled as in **c**. **f**, **g** Frequency of reversed replication forks in U2OS cells transfected with control siRNA (Ctrl) or with siRNAs targeting *RAD51C*, *RAD51D*, or *XRCC3* for 48 h and treated with 50 nM CPT (1 h). **f** Electron micrograph of a representative reversed replication fork from CPT-treated U2OS cells (P parental duplex, D daughter duplexes, R regressed arm; scale-bar, 100 nm). **g** Graph-bar showing mean and SD from three independent EM experiments. In brackets, total number of molecules analyzed per condition. Statistical analysis: one-way ANOVA followed by Bonferroni test; ns not significant; ***$p$-value = 0.0004. Source data for **b**–**g** are provided in the Source Data file.

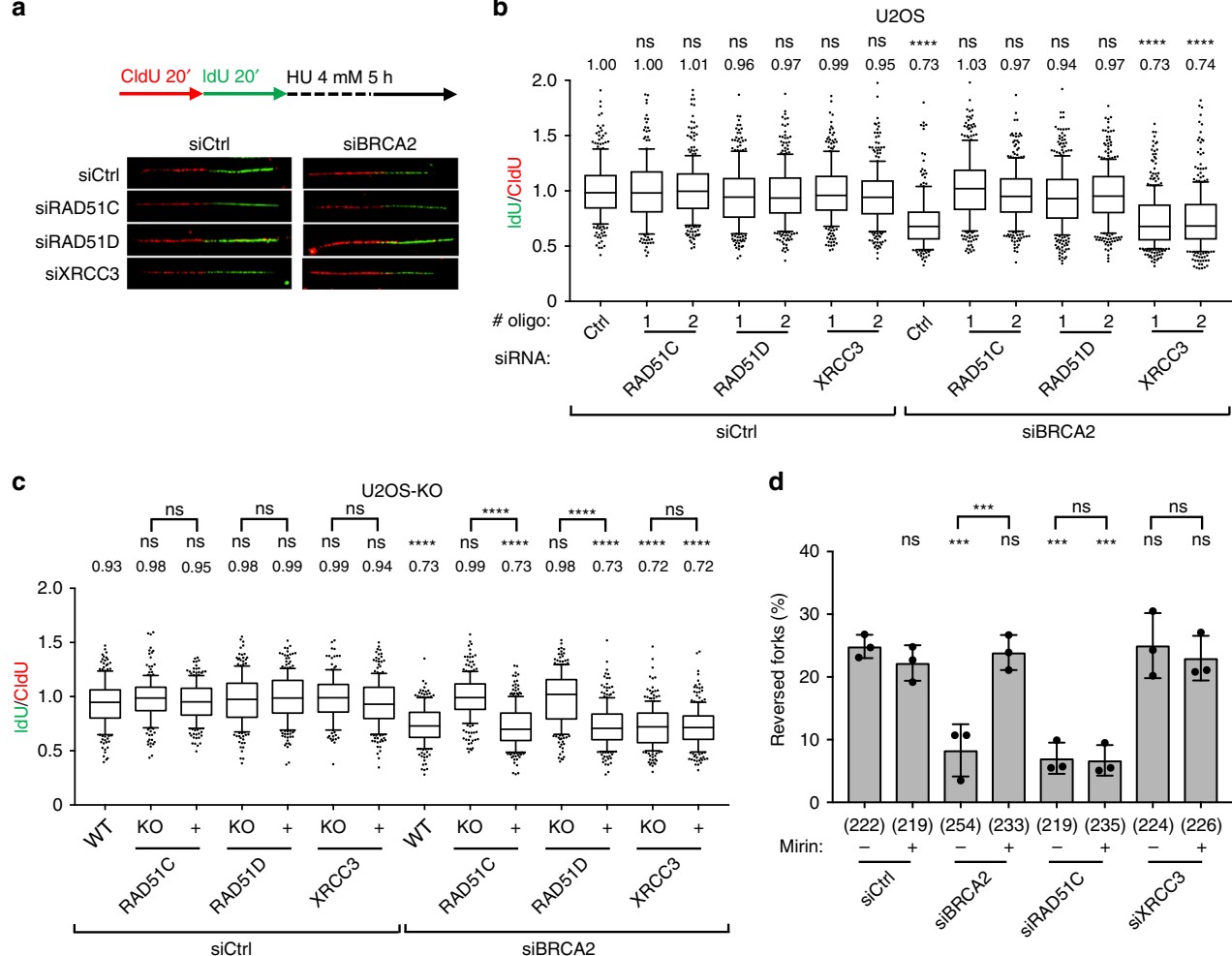

**Fig. 3 BCDX2 complex, but not CX3, promotes reversed fork degradation in BRCA2-depleted U2OS cells. a**, **b** DNA fiber analysis of U2OS cells double transfected with a control siRNA (Ctrl) or with siRNAs targeting *RAD51C*, *RAD51D*, and *XRCC3* for 60 h and with BRCA2 siRNA for 48 h. **a** Top: schematic CldU/IdU pulse-labeling protocol to evaluate fork degradation upon HU treatment (4 mM, 5 h). Bottom: representative DNA fibers from each genetic condition. **b** IdU/CldU ratio is plotted as a readout of fork degradation. In **b** and **c**, the numbers indicate the mean value; a minimum of 150 forks was scored in two independent experiments yielding similar results. Bounds of box are 25–75th percentile, center shows the median, whiskers indicate the 10–90 percentiles, data outside this range are drawn as individual dots. Statistical analysis: Kruskal–Wallis test; ns not significant; ****p-value < 0.0001. **c** DNA fiber analysis of KO and reconstituted (+) U2OS cells transfected with a control siRNA (Ctrl) or with BRCA2 siRNA for 48 h and consecutively labeled and HU-treated as in **a**. IdU/CldU ratio is plotted as a readout of fork degradation. **d** Frequency of reversed replication forks in U2OS cells transfected with control siRNA (Ctrl) or with siRNAs targeting *BRCA2*, *RAD51C*, or *XRCC3* for 48 h and treated for 5 h with HU 4 mM; where indicated 50 μM Mirin was added 1 h before HU treatment (6 h total treatment). Graph-bar depicts mean and SD from three independent EM experiments. In brackets, total number of molecules analyzed per condition. Statistical analysis: one-way ANOVA followed by Bonferroni test; ns not significant; ***p-value = 0.0003. Source data for **a**–**d** are provided in the Source Data file.

DNA translocases[5,12,65]. Using an established EM method to visualize replication intermediates[66], we indeed confirmed that downregulation of *RAD51C* and *RAD51D*, but not of *XRCC3*, markedly impairs CPT-induced replication fork reversal (Fig. 2g, Supplementary Table 1a).

**BCDX2 primes stalled fork degradation in BRCA2-defective cells.** We then used the same validated cellular systems to further investigate the role of BCDX2 and CX3 in fork stability, using an established DNA fiber protocol where double-labeled ongoing forks experience prolonged stalling by HU-induced nucleotide depletion (Fig. 3a)[14,17]. As reported, BRCA2-depleted cells displayed nascent DNA degradation at stalled forks, monitored as reduced ratio between second (IdU) and first (CldU) track length (Fig. 3a, b). Despite effective depletion of protein levels (Supplementary Fig. 3a)—which were sufficient to induce

unrestrained fork progression in CPT (Fig. 2c)—downregulation of none of the tested RAD51 paralogs led to detectable fork degradation (Fig. 3a, b). However, downregulation of *RAD51C* or *RAD51D*—but not of *XRCC3*—with two different siRNA sequences fully restored nascent DNA stability in BRCA2-depleted cells (Fig. 3a, b and Supplementary Fig. 3a). Very similar results were obtained in *RAD51C-KO* and in *RAD51D-KO* U2OS cells, and re-expression of the missing protein restored the fork degradation phenotype upon BRCA2 depletion (Fig. 3c). Also in this cellular system, genetic inactivation of *XRCC3* had no detectable effects on nascent DNA stability, in either BRCA2-proficient or -deficient cells (Fig. 3c). Consistently, downregulation of *RAD51C* or *RAD51D*—but not *XRCC3*—fully restored stalled fork stability also in BRCA2-depleted RPE-1 cells (Supplementary Fig. 3b–c). On one hand, these observations suggest that inactivation of any of the RAD51 paralogs does not

affect stalled fork integrity in human cells. On the other hand, in light of the data in Fig. 2, they also suggest that BCDX2—but not CX3—mediates the formation of reversed forks, which serve then as entry points for nucleolytic degradation of nascent DNA in BRCA2-defective cells. EM visualization confirmed this hypothesis: as reported in different cellular systems[13,14,22], BRCA2-depleted U2OS cells displayed a marked reduction in HU-induced reversed fork frequency, which was readily suppressed by MRE11 inhibition using mirin (Fig. 3d, Supplementary Table 1b). Conversely, mirin did not restore reversed fork frequency upon RAD51C depletion and XRCC3 depletion had no significant effects (Fig. 3d, Supplementary Table 1b). These EM data on RAD51C closely resemble previous observations obtained by RAD51 depletion[14] and, combined with the data in Fig. 3b–c, strongly suggest that BCDX2—but not CX3—is required for fork reversal, upstream of BRCA2-mediated stabilization of reversed forks.

**CX3 mediates efficient restart of reversed forks after stalling**. We next tested by DNA fiber spreading whether RAD51 paralogs assist the restart of stalled replication forks, measuring number and length of replicated tracks after HU removal (Fig. 4a). In our experimental conditions (24 h of siRNA-mediated downregulation), *RAD51* inactivation did not affect the efficiency of fork restart and slightly increased the velocity of restarting forks (Fig. 4a and Supplementary Fig. 4a). This likely reflects impaired fork remodeling associated with RAD51 depletion, as non-reversed forks may be faster in restarting DNA synthesis once nucleotide levels are restored, possibly by efficient replication fork repriming events. Accordingly, downregulation of *RAD51C* or *RAD51D*—which are also required for fork reversal (Figs. 2 and 3)—led to similar mild effects on fork restart (Fig. 4a and Supplementary Fig. 4a). Conversely, XRCC3 depletion drastically increased the fraction of forks failing to restart DNA synthesis and delayed progression of the restarting forks (Fig. 4a and Supplementary Fig. 4a). Very similar effects on efficiency and velocity of fork restart were obtained in U2OS cells carrying genetic ablation of these genes and all effects were fully complemented by re-expression of the missing protein (Supplementary Fig. 4b–c). As XRCC3 is not required for fork reversal (Figs. 2 and 3), we reasoned that XRCC3-depleted cells might be specifically impaired in their ability to restart forks that had previously been reversed. Indeed, co-depletion of RAD51 in XRCC3-depleted U2OS cells fully rescued efficiency and velocity of fork restart (Fig. 4b and Supplementary Fig. 4d). Similarly, XRCC3 depletion impaired fork restart in ZRANB3-proficient, but not in ZRANB3-deficient cells (Fig. 4c and Supplementary Fig. 4e). Moreover, also RAD51D depletion rescued fork restart efficiency and velocity in XRCC3-depleted cells (Fig. 4d, Supplementary Fig. 4f), further supporting the role of BCDX2 in fork reversal, upstream of XRCC3-mediated restart. Finally, as XRCC3 is destabilized in RAD51C-depleted cells (Fig. 2b), *RAD51C* downregulation itself represents a third condition that impairs fork reversal and rescues fork restart in the absence of XRCC3 (Fig. 4a). Overall, these data establish a specific role for the CX3 complex in replication fork restart, downstream of RAD51-, BCDX2-, and ZRANB3-dependent fork reversal.

**BCDX2 fuels chromosomal instability in BRCA2-defective cells**. As reversed forks are the entry points for unscheduled fork degradation in BRCA2-depleted cells, we finally tested whether genetic impairment of *RAD51C* and *RAD51D*—which are required for fork reversal—may restore genome stability in BRCA2-defective cells. In metaphase spreads, depletion of RAD51 paralogs *per se* did not detectably increase chromosomal abnormalities (Fig. 5a). However, depletion of RAD51C or

RAD51D—but not of XRCC3—suppressed the chromosomal instability associated with BRCA2 depletion (Fig. 5a), suggesting that efficient fork remodeling by BCDX2 engages BRCA2 in a crucial genome maintenance function.

**Discussion**
The recent discoveries that mutations in classical RAD51 paralog genes predispose to cancer and Fanconi anemia[47–53] encouraged new mechanistic studies on these HR accessory factors, particularly focused on the human proteins and their possible role in the replication stress response. Human RAD51 paralogs were first expressed and purified over two decades ago, leading to the identification of two main complexes: BCDX2 (composed of RAD51B, RAD51C, RAD51D, XRCC2) and CX3 (composed of RAD51C and XRCC3). However, biochemical insights have remained relatively scarce to date. Furthermore, until very recently, genetic investigations on these proteins were also largely limited to single siRNAs or mutated cell lines in specific model systems, calling for more systematic genetic investigations of these proteins in isogenic human systems, particularly with respect to their role upon replication stress.

Here, we selectively inactivated the BCDX2 complex (via *RAD51D* inactivation), the CX3 complex (via *XRCC3* inactivation), or both complexes (via *RAD51C* inactivation) in U2OS and RPE-1 human cells, and performed single-molecule investigations on replication intermediates. We show that the BCDX2 complex drives reversed replication fork formation, presumably by assisting the central recombinase RAD51, which was previously shown to mediate this step of fork remodeling[5,14]. Differently from replication fork protection and HR-mediated DSB repair, reversed fork formation does not require stable RAD51 nucleofilaments[14]. This suggests that BCDX2 may assist DNA translocases and RAD51 in driving parental strand reannealing, while partially replacing RPA on ssDNA stretches accumulating at stalled forks (Fig. 5b). RAD51 paralogs were shown in yeast and *C. elegans* to biochemically remodel preassembled RAD51 nucleofilaments[40–42]. It is possible that RAD51 paralogs enhance and stabilise spontaneous RAD51 monomer binding to short ssDNA stretches at stalled forks, thus supporting the formation of a metastable nucleofilament, with sufficient flexibility to sustain the high torsional constrains at replication forks and promote fork reversal. Alternatively, RAD51 paralogs might stimulate the spontaneous formation of BRCA2-independent short stretches of RAD51 monomers at stalled forks by preventing the action of human antirecombinase activities[40]. However, our data do not exclude other possible RAD51-independent mechanisms by which BCDX2 may foster the reversal reaction, such as intrinsic strand annealing activities[29] or specific interactions with different enzymatic activities (e.g., DNA translocases) and cofactors involved in fork reversal. While most reported biochemical assays of fork reversal were performed in the absence of ssDNA binding proteins, the addition of RPA or RAD51—when tested—had profound effects on the efficiency and/or directionality of these reactions[67,68]. Holistic biochemical reconstitution of these important transactions at the replication fork seems necessary to elucidate the precise molecular mechanism by which BCDX2 and RAD51 collaborate with DNA translocases to drive efficient fork reversal.

As already shown for RAD51, which plays key genetically uncoupled roles both in the formation and in the protection of reversed forks[14,16,17], our data do not exclude that BCDX2 plays also a crucial role in protection and restart of stalled replication forks (Fig. 5b). However, as for RAD51, this role may be masked by its crucial upstream function in replication fork reversal. As exemplified for RAD51 by the T131P separation-of-function mutant[14], it is likely that hypomorphic

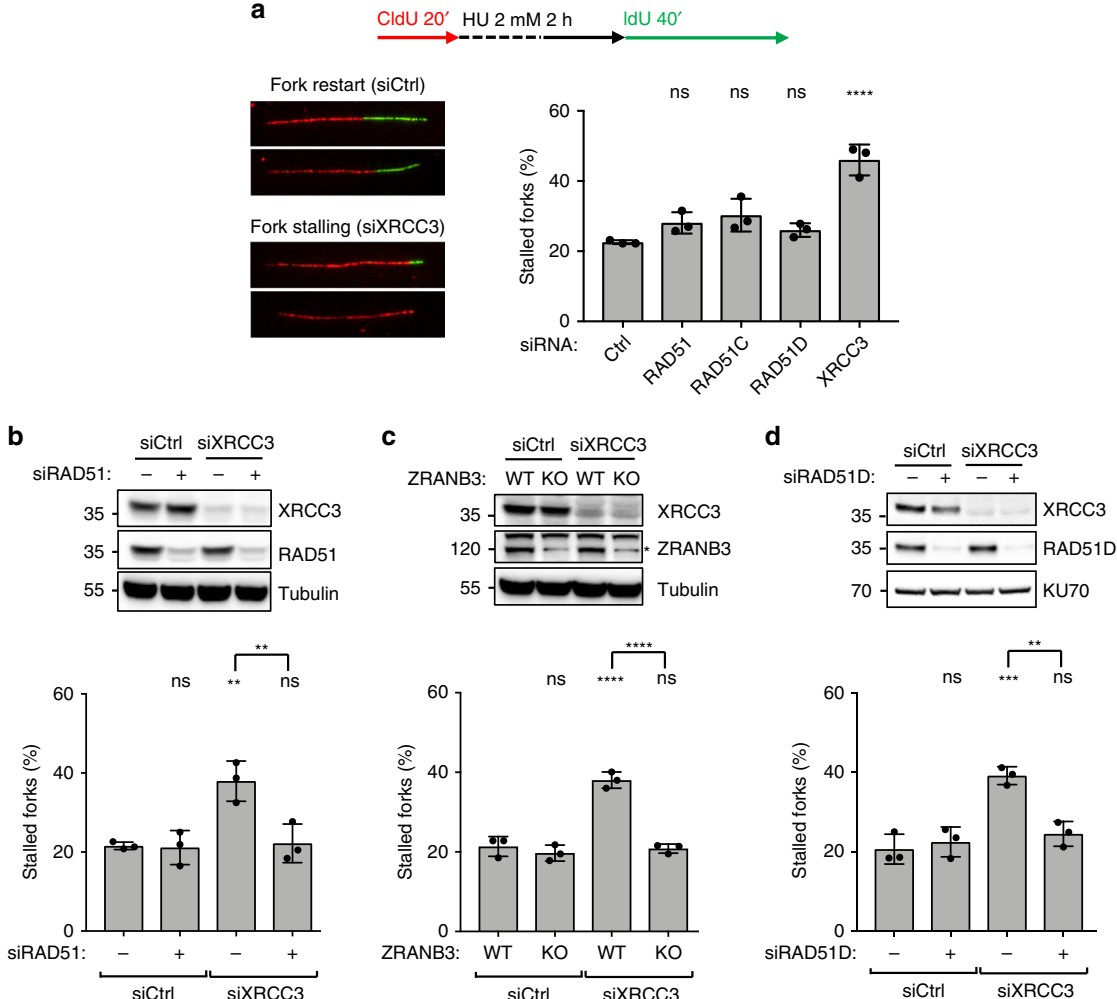

**Fig. 4 CX3 complex promotes reversed fork restart in U2OS cells. a** DNA fiber analysis of U2OS cells transfected for 24–48 h (24 h for *RAD51*, 48 h for all the other genes) with a control siRNA (Ctrl) or with siRNAs targeting *RAD51*, *RAD51C*, *RAD51D*, and *XRCC3* to investigate replication fork restart upon HU. Top: schematic CldU/IdU pulse-labeling protocol to evaluate fork restart upon HU treatment (2 mM, 2 h). Bottom-left: representative stalled and restarting forks. Bottom-right: frequency of fork stalling for each genetic condition. In **a–d**, the graph-bar depicts mean and SD from three independent experiments. Statistical analysis: one-way ANOVA followed by Bonferroni test; ns not significant; ****$p$-value < 0.0001. **b** DNA fiber analysis of U2OS cells double transfected with a control siRNA (Ctrl) or with a siRNA targeting *XRCC3* (48 h) or *RAD51* (24 h) to investigate replication fork restart upon HU treatments; labeling protocol as in **a**. Top: levels of indicated proteins, assessed by western blot; Tubulin, loading control. In **b** and **d** multiple gels/blots were processed in parallel, ensuring equal and comparable loading across gels. Bottom: frequency of fork stalling. ns not significant; **$p$-value = 0.0068 (1 versus 3) and 0.0087 (3 versus 4). **c** DNA fiber analysis of wild-type (WT) or *ZRANB3* Knock-Out (KO) U2OS cells transfected with a control siRNA (Ctrl) or with siRNAs targeting *XRCC3* (48 h) to investigate replication fork restart upon HU treatments; labeling protocol as in **a**. Top: indicated protein levels, assessed by western blot; Tubulin, loading control; asterisk, specific band. Bottom: frequency of fork stalling. ns not significant; ****$p$-value < 0.001. **d** DNA fiber analysis of U2OS cells double transfected with a control siRNA (Ctrl), with siRNAs targeting *XRCC3* (48 h) or *RAD51D* (60 h) to investigate replication fork restart upon HU treatments; labeling protocol as in **a**. Top: indicated protein levels, assessed by western blot; KU70, loading control. Bottom: frequency of fork stalling. ns not significant; **$p$-value = 0.0024; ***$p$-value = 0.0007. Source data for **a–d** are provided in the Source Data file.

conditions of RAD51 paralog inactivation and/or different levels and regulations of these proteins in different model systems may explain different phenotypes in fork progression and degradation upon inactivation of BCDX2 and CX3 components[54–57]. In this context, it will be paramount to use isogenic replacement systems to test fork progression, remodeling and stability with specific RAD51 paralog mutants linked with cancer and Fanconi anemia[47–53], to reveal the specific function(s) of these proteins in preventing human disease.

Our data uncover a role for the CX3 complex in the restart of reversed replication forks, establishing a sequential engagement of different RAD51 paralog complexes in stalled fork remodeling and reactivation (Fig. 5b). Interestingly, a downstream role for

CX3—with respect to BCDX2—had been hypothesized also in classical HR-mediated repair of DSBs, based on the limited requirement of XRCC3 for RAD51 foci formation[45] and on the Holliday junction resolution activity found associated with the CX3 complex[69]. Recent data with *CRISPR-KO* cells challenge this model[46], but may also reflect long-term effects of chronic *XRCC3* inactivation and cell adaptation mechanisms to avoid the resultant accumulation of toxic recombination intermediates. While we did not succeed in monitoring directly the recruitment of RAD51 paralogs to replication forks with currently available tools, XRCC3 was previously shown to bind nascent DNA with a delayed kinetic compared to BCDX2 subunits[54], which is consistent with the sequential model proposed here (Fig. 5b).

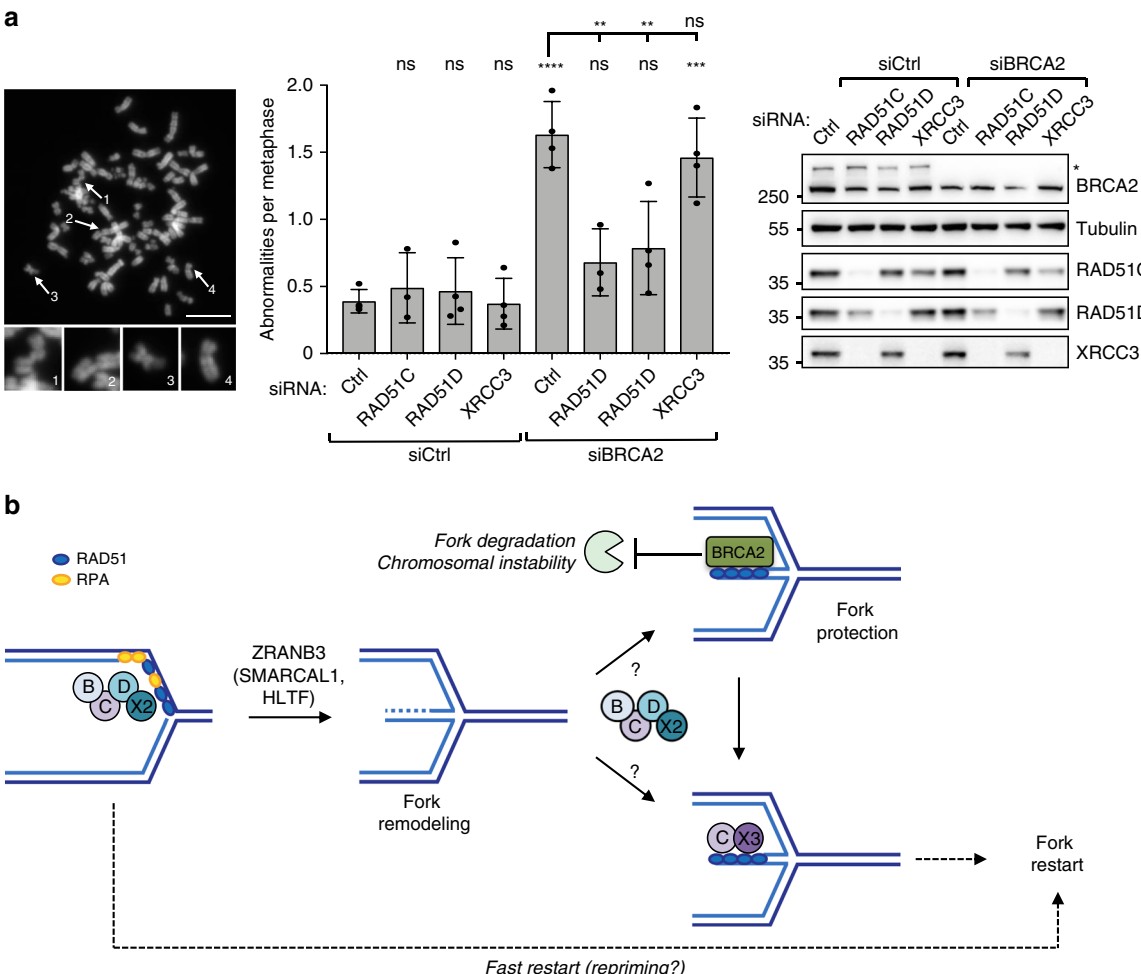

**Fig. 5 BCDX2, but not CX3, promotes chromosomal instability in BRCA2-defective cells. a** Metaphase spread analysis for detecting chromosomal aberrations in U2OS cells double transfected with a control siRNA (Ctrl) or with siRNAs targeting *RAD51C*, *RAD51D*, and *XRCC3* for 60 h and with *BRCA2* siRNA for 48 h and treated with 4 mM HU for 5 hr. Left: representative metaphase spread image (scale-bar, 5 μm); 1, 2, 3 representative breaks and 4 representative intact chromosome. Center: number of chromosomal abnormalities for each genetic condition. The graph-bar depicts mean and SD from at least three independent experiments. Statistical analysis: one-way ANOVA followed by Bonferroni test; ns not significant; **$p$-value $= 0.0012$ (5 versus 6) and 0.0020 (5 versus 7); ***$p$-value $= 0.0001$; ****$p$-value $< 0.0001$. Right: levels of indicated proteins, assessed by western blot; Tubulin, loading control; asterisk, specific band. Multiple gels/blots were processed in parallel, ensuring equal and comparable loading across gels. Source data are provided in the Source Data file. **b** Proposed model for the sequential role of BCDX2 and CX3 paralog complexes in replication fork remodeling, protection and restart: the BCDX2 complex helps RAD51 and ZRANB3 in driving fork remodeling. During fork stalling, regressed arms are protected from unscheduled nucleolytic degradation by BRCA2-mediated stabilization of RAD51 nucleofilament. Preventing fork remodeling by BCDX2 inactivation suppresses genetic instability of BRCA2-deficient cells upon sustained replicative stress. Similarly to RAD51, BCDX2 may be involved in fork protection (and possibly fork restart), but this role may be genetically masked by its upstream role in promoting fork reversal. Once DNA synthesis can resume, the CX3 complex promotes reversed fork restart, likely by engaging RAD51-bound regressed arms in efficient strand invasion events. In absence of fork remodeling, stalled forks undergo a fast, but possibly inaccurate restart mode, likely driven by repriming events.

Although both BRCA2 and XRCC3 were shown to play a role downstream of fork remodeling, they display rather different, yet possibly coordinated reversal-dependent functions. While *BRCA2* defects lead to reversed fork degradation, but mild fork restart defects[13,17,20], *XRCC3* inactivation does not impair fork stability, but severely impairs single-fork reactivation once the stalling agent has been removed (Fig. 4), thus respectively handling a static (fork protection) and dynamic (fork restart) replication stress response. While the role of BRCA2 in fork protection is clearly linked to RAD51 nucleofilament stabilization on the regressed arm[14,17], we propose that XRCC3 assists fork restart by promoting RAD51-mediated strand invasion of parental DNA by the regressed arm, which may be important for at least a subset of fork restart events. Consistently, the human CX3 complex is specifically observed to promote RAD51 nucleofilament remodeling and stability, thereby

promoting strand invasion (Lumir Krejci, personal communication), similarly to what was previously reported for the unique RAD51 paralogs complex in *C. elegans*[41]. Alternatively, CX3 may facilitate the recruitment of specialized factors, catalyzing fork restoration via branch migration or controlled fork processing. Overall, several different scenarios could be envisioned to explain CX3 contribution to fork restart, and only a complete biochemical reconstitution of this reaction in the future—including known fork restart activities—will possibly clarify.

Importantly, BCDX2 inactivation, but not *XRCC3* down-regulation, rescued chromosomal abnormalities upon fork stalling in BRCA2-defective cells. These data are aligned with previous findings with DNA translocase inactivation[22] and reinforce the concept of replication fork reversal as a double-edged sword in genome maintenance, requiring tight biological control. While on

one hand fork reversal represents an important mechanism to avoid replication stress-associated chromosomal breakage[5,8,58], it can also drive clinically relevant pathogenic transactions in the absence of functional fork protection and restart mechanisms[10,13,20,22]. Such data suggest that multiple defects in the same remodeling pathway may occasionally combine, leading to different outcomes in terms of genome stability. As reversal-dependent degradation was shown to be a major molecular determinant of chemosensitivity and -resistance in HR-defective tumors[20], it will be essential to test the molecular phenotypes upon cancer-associated mutations in RAD51 paralogs, and thereby to possibly establish predictive parameters for chemo-sensitivity in this subset of HR-defective tumors.

## Methods

**Cell culture and cell lines**. Human U2OS and hTERT-RPE-1 cells were grown in Dulbecco's modified Eagle's medium (DMEM) supplemented with 10% fetal bovine serum (FBS; GIBCO/Thermo Fisher Scientific) and 1% penicillin-streptomycin antibiotics (100 U/mL penicillin and 100 µg/mL streptomycin) under standard cell culture conditions (humidified atmosphere, 6% $CO_2$) at 37 °C.

ZRANB3-proficient and Knock-Out U2OS cells were kindly provided by Dr. David Cortez.

RAD51 paralog CRISPR-Cas9-based Knock-Out U2OS cells have been generated and genetically characterized as recently reported[46].

**siRNA transfection and sequences**. The QIBC-based siRNA screen was per-formed by reverse transfection of U2OS cells in CELLSTAR 96-well plates (Greiner Bio-One) for 48 h at a cell density of 5000 cells per well at the time of transfection with Ambion Silencer Select siRNAs at a final concentration of 5 nM using HiPerFect reagent (Qiagen).

Individual siRNA transfections were performed using Lipofectamine RNAiMAX (ThermoFisher Scientific) at a concentration of 10 nM (Ambion Silencer Select) or 40 nM (Microsynth AG) according to manufacturer's instruction and experiments performed between 48 and 72 h post transfection as indicated.

The oligonucleotides used for individual assays in this study are presented in Supplementary Table 2. Unless stated otherwise in the Figure legend siCtrl (as negative siRNA control), siRAD51C #2, siRAD51D #1, and XRCC3 #1 have been used.

**Drugs and reagents**. The following reagents were used to treat the cells for the indicated time at the indicated final concentrations before collection: Hydroxyurea (HU; H8627, Sigma–Aldrich) was prepared in double-distilled $H_2O$ to obtain a 1 M stock and stored at −20 °C; Mirin (M9948, Sigma–Aldrich) was dissolved in DMSO to yield a 50 mM stock and aliquots were stored at −80 °C; Camptothecin (CPT; C9911, Sigma–Aldrich) was dissolved in DMSO to yield a 5 mM stock (freshly made); Nocodazole (M1404, Sigma–Aldrich) was prepared in DMSO at the final concentration of 1 mg ml$^{-1}$, and aliquots were stored at −20 °C.

**Immunoblotting**. Cell were washed in cold PBS and lysed in NP-40 buffer (50 mM Tris-HCl [pH 7.5], 150 mM NaCl, 1% NP-40) supplemented with 1× protease inhibitor cocktail (cOmplete, Roche) and phosphatase inhibitors (20 mM NaF, 1 mM $Na_3VO_4$ and 5 mM $Na_4P_2O_7$). Cell extract were diluted with NuPAGE LDS sample buffer (Thermo Fisher Scientific) containing DTT and heated at 60 °C for 10 min. Proteins, together with the PageRuler Plus Prestained Protein Ladder (Thermo Fisher Scientific, 26620), were resolved on NuPAGE 4–12%, Bis-Tris or 3–8% Tris-Acetate gels (Thermo Fisher Scientific) using MOPS and Tris-Acetate SDS running buffers respectively, and transferred on nitrocellulose membranes. Membranes were blocked with TBS containing Tween-20 (0.1%) and 5% milk powder for 1 h at room temperature (RT), incubated with the indicated primary antibodies overnight at 4 °C and secondary antibodies for 1 h at RT. Protein were visualized using WesternBright ECL reagent (Advansta) and the Fusion Solo S imaging system (Vilber Lourmat). Uncropped and unprocessed scans of each blot are provided in the Source Data file.

The following primary antibodies were used: RAD51B mouse (sc-377192, Santa Cruz Biotechnology, 1:200), RAD51C rabbit (ab95069, Abcam, 1:2000), RAD51D rabbit (ab202063, Abcam, 1:500), XRCC2 mouse (sc-365854, Santa Cruz Biotechnology, 1:200), XRCC3 mouse (sc-271714, Santa Cruz Biotechnology, 1:200), BRCA2 mouse (OP-95, EMD Millipore, 1:500), RAD51 rabbit (Bioacademia 70-002, 1:5000), ZRANB3 rabbit (23111-1-AP, Proteintech, 1:500), KU70 mouse (ab202022, Abcam, 1:2000), and α-Tubulin mouse (T9026, Sigma–Aldrich, 1:10000).

**Immunostaining**. Cells were grown on glass coverslips or 96-well plates, washed in PBS, pre-extracted for 5 min at 4 °C in CSK buffer (10 mM Hepes-KOH [pH 7.4], 300 mM sucrose, 50 mM NaCl, 3 mM $MgCl_2$, 1 mM EGTA, 0.5% Triton X-100), fixed in 4% paraformaldehyde in PBS for 12 min at RT, washed twice in PBS,

permeabilized in PBS supplemented with 0.3% Triton X-100, washed twice in PBS and blocked with 3% BSA in PBS twice for 15 min. Rabbit polyclonal RAD51 antibody (Bioacademia 70-002, 1:2000) and secondary antibody (Alexa fluor-ophores, Life Technologies) were diluted in 1.5% BSA in PBS and incubation were, respectively, performed for 2 or 1 h at RT. Coverslips were washed with PBS supplemented with 0.05% Tween-20 and incubated for 10 min with PBS containing 4′,6-Diamidino-2-Phenylindole Dihydrochloride (DAPI, 0.5 µg/mL) at RT to stain DNA. Following three washing steps in PBS, coverslips were rinsed with distilled water, air-dried and mounted with ProLong Gold AntiFade (Thermo Fisher Sci-entific). When indicated the Click-iT EdU Alexa Fluor Imaging Kit (Thermo Fisher Scientific) was used for EdU detection according to manufacturer's instruction.

**Quantitative image-based cytometry (QIBC)**. QIBC experiments were per-formed on an Olympus ScanR Screening System equipped with an inverted motorized Olympus IX83 microscope, a motorized stage, IR-laser hardware autofocus, a fast emission filter wheel with single band emission filters, and a digital monochrome Hamamatsu ORCA-FLASH 4.0 V2 sCMOS camera (2048 × 2048 pixel, pixel size 6.5 × 6.5 µm, 12 bit dynamics), as described previously[70,71]. For each condition, image information of large cohorts of cells (typically at least 500 cells for the UPLSAPO ×40 objective (NA 0.9) and at least 2000 cells for the UPLSAPO ×20 objective (NA 0.75)) was acquired under non-saturating condi-tions. Identical settings were applied to all samples within one experiment. Images were analyzed with the Olympus ScanR Image Analysis Software, a dynamic background correction was applied, nuclei segmentation was performed using an integrated intensity-based object detection module using the DAPI signal, and foci segmentation was performed using an integrated spot-detection module. All downstream analyses were focused on properly detected nuclei containing a 2C-4C DNA content as measured by total and mean DAPI intensities. Fluorescence intensities were quantified and are depicted as arbitrary units. Color-coded scat-terplots of asynchronous cell populations were generated with Spotfire data visualization software (TIBCO). Within one experiment, similar cell numbers were compared for the different conditions. For visualizing discrete data in scatterplots, mild jittering (random displacement of data points along the discrete data axes) was applied in order to demerge overlapping data points. For the siRNA screen, genes were ranked according to a z-score ($z = (x − \mu)/\sigma$ with x being the mean number of RAD51 foci per cell for each knockdown, µ being the mean number of RAD51 foci per cell across all conditions, and σ being the standard deviation of the mean number of RAD51 foci per cell across all conditions). Representative scat-terplots and quantifications of independent experiments are shown. Representative images, in which the individual color channels have been adjusted for brightness and contrast, accompany selected quantifications.

**DNA fiber spreading analysis**. Asynchronously growing cells were labeled with the thymidine analogues 5-Chloro-2′-deoxyuridine (CldU, 30 µM), washed 3 times with PBS, followed by 5-Iodo-2′-deoxyuridine (IdU, 250 µM) and treated with HU and CPT as indicated[58]. The cells were quickly trypsinized and resuspended in ice-cold PBS at $2.5 × 10^5$ cells per ml. The labeled cells were diluted 1:1 with unlabeled cells, and 3 µl of cells were mixed with 7.5 µl of lysis buffer (200 mM Tris-HCl [pH 7.5], 50 mM EDTA, 0.5% (w/v) SDS) on a glass slide. After 9 min, the slides were tilted at 15°–45°, and the resulting DNA spreads were air-dried, fixed in 3:1 methanol/acetic acid overnight at 4 °C. The DNA fibers were denatured with 2.5 M HCl for 1 h, washed with PBS and blocked with 2% BSA in PBS supplemented with 0.1% Tween-20 for 40 min. The newly replicated CldU and IdU tracks were labeled (for 2.5 h in the dark, at RT) with anti-BrdU/CldU antibodies recognizing CldU (ab6326, Abcam, rat, 1:500) and BrdU/IdU (347580, Becton Dickinson, mouse, 1:100), respectively. After washing 5 × 3 min in PBS supplemented with 0.2% Tween-20, the following sec-ondary antibodies were used (incubated for 2 h in the dark, at RT): anti-mouse Alexa 488 (Molecular Probes, 1:300), anti-rat Cy3 (Jackson Immunoresearch, 1:150). After washing 5 × 3 min each in PBS supplemented with 0.2% Tween-20 the slides were air-dried completely and mounted with 20 uL/slide ProLong Gold AntiFade (Thermo Fisher Scientific). Images were acquired using an Olympus IX81 fluorescence microscope equipped with a CCD camera (Orca AG, Hamamatsu). CldU and IdU tract lengths were measured using the line tool in ImageJ64 software. In all the fork restart experiments (Fig. 4), defective fork restart (i.e., fork stalling) was defined as a ratio (length of green vs red) <0.1.

**Electron microscopic analysis of genomic DNA**. Following the depletion of the protein of interest, asynchronous subconfluent cells were treated with 50 nM CPT for 1 h or 4 mM HU for 5 h. Where indicated, cells were pretreated with 50 µM Mirin for 1 h. Cells were collected, resuspended in ice-cold PBS and crosslinked with 4,5′, 8-trimethylpsoralen (10 µg/ml final concentration), followed by irradiation pulses with UV 365 nm monochromatic light (UV Stratalinker 1800; Agilent Technologies). For DNA extraction[66], cells were lysed (1.28 M sucrose, 40 mM Tris-HCl [pH 7.5], 20 mM $MgCl_2$, and 4% Triton X-100; Qiagen) and digested (800 mM guanidine–HCl, 30 mM Tris-HCl [pH 8.0], 30 mM EDTA [pH 8.0], 5% Tween-20, and 0.5% Triton X-100) at 50 °C for 2 h in presence of 1 mg/ml proteinase K. The DNA was purified using chloroform/isoamylalcohol (24:1) and precipitated in 0.7 volume of iso-propanol. Finally, the DNA was washed with 70% EtOH and resuspended in 200 µl TE (Tris-EDTA) buffer. 100 U of restriction enzyme (PvuII high fidelity, New

England Biolabs) were used to digest 6 µg of mammalian genomic DNA for 5 h. RNase A (Sigma–Aldrich, R5503) to a final concentration of 250 ug/ml was added for the last 2 h of this incubation. The digested DNA was transferred into Microcon DNA Fast Flow centrifugal filters (Merck MRCF0R100) and washed two times with 200 ul TE. Ultimately, the digested DNA was concentrated and recovered by inverting the filters. The Benzyldimethylalkylammonium chloride (BAC) method was used to spread the DNA on the water surface and then load it on carbon-coated 400-mesh nickel grids (G2400N, Plano Gmbh). Subsequently, DNA was coated with platinum using a High Vacuum Evaporator BAF060 (Leica). The grids were scanned using a transmission electron microscope (Tecnai G2 Spirit; FEI; LaB6 filament; high tension ≤120 kV) and pictures were acquired with a side mount charge-coupled device camera (2600 × 4000 pixels; Orius 1000; Gatan, Inc.) and processed with DigitalMicrograph Version 1.83.842 (Gatan, Inc.). For each experimental condition at least 70 replication fork molecules were analyzed in three different biological replicates by using ImageJ64 .

**Chromosomal breakage and abnormalities by metaphase spreading**. After transfection with the indicated siRNAs, cells were treated for 5 h with 4 mM HU. The compound was washed off three times with 1× PBS, upon which cells were released into fresh medium containing 200 ng/ml nocodazole for 16 h. Cells were harvested and swollen with 75 mM KCl for 20 min at 37 °C. Swollen mitotic cells were collected and fixed with methanol:acetic acid (3:1). The fixing step was repeated two times. Fixed cells were dropped onto prehydrated glass slides and air-dried overnight. The following day, slides were mounted with Vectashield medium containing DAPI. Microscopy was performed on a Leica DM6 B upright digital research microscope equipped with a DFC360 FX Leica camera. Images were analyzed using ImageJ64 and visible chromatid breaks/gaps were counted. For each experimental condition at least 50 metaphases were analyzed in three different biological replicates.

**Statistical analysis**. For QIBC analysis, between 9 and 15 images per condition, depending on the microscope objective used and the cell confluence were acquired in an unbiased fashion from asynchronous cell populations grown on glass coverslips or multi-well plates. Typically, between 2000 and 5000 cells per condition were analyzed and representative single-cell data of cell cohorts of comparable size are shown as two-dimensional cell-cycle-resolved or one-dimensional scatterplots.

For DNA fiber length measurements, at least 150 fibers were scored for each condition and every experiment was repeated at least twice. The results are depicted as median plus 10–90 percentile whisker plots and Kurskal–Wallis test was used for statistical analysis.

In all the other experiments, including DNA fiber analysis of fork restart, the statistical significance for three different biological replicates was determined by one-way ANOVA followed by Bonferroni test.

GraphPad Prism7 for MacOSX was used for all statistical analyses.

**Reporting summary**. Further information on research design is available in the Nature Research Reporting Summary linked to this article.

## Data availability

Source data are provided with this paper. All other data that support the findings of this study are available from the authors upon reasonable request.

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

## Acknowledgements

We thank the Center for Microscopy and Image Analysis of the University of Zurich for assistance with microscopy and imaging analysis. We are grateful to Eli Rothenberg, Huijun Xue, Petr Cejka, Swagata Halder and members of the Lopes and Altmeyer labs for technical assistance and fruitful discussions, and to Jean-Yves Masson for critical reading of the manuscript. Work in the Lopes lab was supported by the SNF grant 31003A_169959, the ERC Consolidator Grant 617102 and the Swiss Cancer League grant KFS-3967-08-2016 to M.L. M.B. was also supported by the H2020 Marie Sklodowska-Curie postdoctoral fellowship 704817. Work in the Altmeyer lab was supported by SNF grants PP00P3_150690 and PP00P3_179057 and the European Research Council (ERC) under the European Union's Horizon 2020 research and innovation program ERC Starting Grant 714326 to M.A. F.T. was also supported by a Candoc fellowship from UZH. S.G and M.M. were supported by CRCM state subsidies from CNRS, Inserm, Institute Paoli-Calmettes and Aix-Marseille University, and the French National League against Cancer. E.B.G. was supported by Postdoctoral fellowships from the French National League against Cancer and from Aix-Marseille University Foundation.

## Author contributions

M.B. and F.T. designed and performed the QIBC-based screen and conducted validation experiments; M.B., S.M., and J.K. performed most DNA fiber assays, while M.S.U. and S.M. performed most EM experiments, assisted by M.B., J.F., J.K., J.A.S., and M.D.P.; J.K. and J.A.S. performed the metaphase spread analysis. E.B.G., S.G., and M.M. produced all CRISPR-KO cell lines and shared them ahead of publication. M.L. designed the project and wrote the first manuscript draft, helped by M.B., F.T., and M.A.; M.L. and M.A. supervised the project; all authors read and provided feedback on the manuscript.

## Competing interests

The authors declare no competing interests.
