## [Peer Review File · Nature Communications]

Reviewers' comments:

Reviewer #1 (Remarks to the Author):

This manuscript describes the results of a very well conducted study focused on detailing the molecular mechanisms by which the classical 5 RAD51 paralogs affect replication fork remodeling and protection. The major findings are that the 2 main RAD51 paralog complexes orchestrate fork protection sequentially. The investigators show that BCDX2 functions upstream of C/X3 (and also of BRCA2) and assists with fork reversal, i.e. mediating fork slowing upon genotoxic stress, and stalled fork degradation in BRCA2-defective cells. CX3 was found to be dispensable for fork reversal, but is needed for the efficient restart of reversed forks.

The manuscript addresses a very important question in the field, which is how RAD51-mediated fork reversal is catalyzed and which proteins are involved in the regulation of this process. The manuscript is very well written and the limitations of the study are discussed appropriately.

The strengths of the study are that isogenic human cell lines were used. Moreover, further validation experiments were conducted in an untransformed human cell line. Observed phenotypes were challenged by tests for antagonistic effects when compared to known factors involved in replication fork remodeling (e.g., ZRANB3 knockdown).

I have three minor comments:

1. Reference #53 seems inappropriately cited as this reference deals with the effects of PARP-1 on forks (not with a paralog).
2. Somyajit K et al. NAR (2015), already showed that XRCC2 is dispensable for fork restart and that X3/C promote fork restart. The investigators cite this paper and state in this context – "...that the functional role of these factors in replication has remained elusive." I suggest that this statement be omitted or toned down (also given the following sentence which again cites ref #51).
3. RAD51 downregulation would be expected to negatively impact fork restart, but does not do so after 2 mM HU for 2h. Under these conditions, forks restart faster when RAD51 is depleted. Is it expected that the restart of these stalled forks does not require any RAD51, or is the remaining RAD51 sufficient to do so?

Reviewer #2 (Remarks to the Author):

In the manuscript by Berti M. and Teloni F. et al., the authors identified the human RAD51 paralogs (RAD51B, RAD51C, RAD51D, XRCC2 and XRCC3) from a targeted siRNA library in U2OS and RPE-1 cells under replication stress. While previous work has investigated the role of the RAD51 paralogs in canonical DSB repair, little cellular investigation has been performed in other settings of genome stability. The authors showed that the BCDX2 components RAD51C and RAD51D enable fork slowing and their depletion results in a lower percentage of reversed forks. Next, the authors created CRISPR KO cells of RAD51C, RAD51D, and XRCC3 and demonstrate in BRCA2 depleted cells that the BCDX2 complex components RAD51C and RAD51D are essential for restoring fork stability. The authors find that XRCC3 depletion increased the percentage of stalled forks and that the CX3 complex is important for replication restart. In conclusion, the authors provide evidence that the RAD51 paralog complexes have specific and distinct functions under conditions of replication fork stress (CPT vs HU). This important insight suggests that the two subcomplexes work sequentially to overcome replication stress. Their model suggests that the RAD51 paralog complex BCDX2 restrains replication progression while the CX3 complex mediates replication restart. These new exciting studies provide insight regarding RAD51 regulation.

Major comments:

- 1) Given the model in which BCDX2 acts early and allows downstream action of CX3 and BRCA2, does loss or depletion of RAD51B, RAD51D, or XRCC2 result in reduced BRCA2 or XRCC3 at the fork? Perhaps iPOND or foci formation upon replication stress would more directly demonstrate this model.
- 2) In the model and introduction, the authors should include the recent description of RAD52 and SMARCAL1 in fork protection (nascent strand) and restart (Malacaria et al., Nature Communications 2019). This is another important example of an HR protein playing noncanonical HR functions. Moreover, the function of RAD52-SMARCAL1 becomes increasingly more important upon BRCA2 deficiency and should be included in the model.
- 3) While under consideration for publication, characterization of the human RAD51 paralog containing complex, the Shu complex, was published (Martino J. et al., Nucleic Acids Research 2019) and is referenced several times throughout the text. Please modify the sentence, "Dedicated studies will be required to explore a possible role for the Shu complex components SWSAP1 and SWS1 in the replication stress response" to include their published role in replication restart (Martino J. et al., Nucleic Acids Research 2019).
- 4) Explicitly state the cell line used for the siRNA screening in your description of Figure 1.
- 5) It is confusing that the authors state that DSBs are not formed under 50 nM CPT treatment but then state "Impairing fork remodeling activities may not only negatively affect RAD51 binding to forks, but can also induce chromosomal breakage and thereby enhance RAD51 accumulation in foci." Please address for consistency.
- 6) sgSWSAP1 and sgSWS1 cells have been assessed for RAD51 foci formation previously (Martino J. et al., Nucleic Acids Research 2019).
- 7) In Figure 1c, SHFM1 is better known in the literature as DSS1 and has been published as this protein name in association with BRCA2, RPA etc.
- 8) In Figure 1h, XRCC3 forms more RAD51 foci upon CPT and IR treatment relative to the other paralog knockdowns. This does not appear to be specific to its role in replication stress but could be due to a less efficient knockdown or a lower overall protein abundance needed for its cellular function. The siXRCC3 data actually mirrors the siControl. The authors should show this in their CRISPR knockouts as it is one of the hits they interrogate.
- 9) In Figure 2, while the study clearly assessed protein expression, the "stability" was not directly assessed. Either perform cycloheximide chases to demonstrate protein stability or remove "stability" from the description for Figure 2.
- 10) In Figure 4b, if BCDX2 and CX3 are functioning sequentially, then the RAD51 and XRCC3 co-depletion should have the same phenotype as RAD51B, RAD51D, or XRCC2 co-depletion with XRCC3. This experiment would strengthen the evidence for distinctive roles for the subcomplexes.

Minor comment:

- 1) "Besides the possibility of incomplete downregulation, this result may also reflect ..." Is it possible that these factors may not have been identified in your screen as they play a role in persistent fork stalling and/or collapse?

Reviewer #3 (Remarks to the Author):

The manuscript "Sequential role of RAD51 paralog complexes in replication fork remodeling and restart" By Berti et al. describes the roles of the RAD51 paralogues recombination factors (RAD51B, RAD51C, RAD51D, XRCC2 and XRCC3) in DNA replication. These paralogues form two distinct complexes the BCDX2, containing RAD51B-D and XRCC2, and CX3 complex containing RAD51C and XRCC3. The authors nicely show that BCDX2 functions to slow down fork progression and promote fork reversal while CX3 complex is not essential for fork reversal but is important for fork restart.

The experiments and the data presented in the manuscript were done in a thorough manner and the data is very interesting. In order to clarify the manuscript more, I suggest the following corrections:

1. I found the introduction much too long, while it is important to provide a thorough background and describe previous works, the authors should focus on the most relevant previous studies and open questions in the field, rather than providing a too long and elaborated account of the background. I urge the authors to at least cut ~30% of the introduction to allow for a more efficient and fast reading of the paper. For example, the last sentence of the first paragraph in page 4: "In addition to..." in my view is not important.
2. Results section page 7: it is not clear what is the size of the siRNA library, how many proteins were targeted in the screen?
3. Page 7 bottom: Define what is a z-score for the general reader?
4. Page 10 bottom, "Importantly, defective fork slowing..." This is a poor phrasing, please rephrase, faster fork progression as an example.
5. Discussion section, page 15: second paragraph, it's good to remind the reader the nature of the two complexes, BCDX2 composed of and CX3 composed of..., since all the data presented target subunits of these complexes (RAD51C, RAD51D and XRCC3).
6. Page 17 first section. Describing the sequential role of the complexes. It is not clear what are the experimental evidences supporting this sequential role and whether the authors can prove which complex is recruited first and which one later on.
7. Page 17 fork restart, can the authors provide more mechanistic evidences regarding how fork restart is promoted, with respect to replisome recruitment and DNA synthesis?
8. Fig. 1, it is not clear how XRCC3 was identified? Fig. 1f show the levels of RAD51 foci of si XRCC3 relative to si control and the level of foci looks very similar. While the authors refer to the milder effects of XRCC3 downregulation in page 9, it is important to show statistical tests between the samples in 1f to test significance differences between different siRNA samples.
9. Fig. 4a, it is not clear what are the examples of fork restart and fork stalling, from what cell line these examples were taken. In addition, what is the criteria for fork stalling? Specifically, what ratio of green/red signal determine the threshold for counting it as fork stalling.
10. Figure 5b, the model, I think that a better choice of colors can allow the reader to more clearly identify the two complexes.

Amir Aharoni

Reviewers' comments:

Reviewer #1 (Remarks to the Author):

This manuscript describes the results of a very well conducted study focused on detailing the molecular mechanisms by which the classical 5 RAD51 paralogs affect replication fork remodeling and protection. The major findings are that the 2 main RAD51 paralog complexes orchestrate fork protection sequentially. The investigators show that BCDX2 functions upstream of C/X3 (and also of BRCA2) and assists with fork reversal, i.e. mediating fork slowing upon genotoxic stress, and stalled fork degradation in BRCA2-defective cells. CX3 was found to be dispensable for fork reversal, but is needed for the efficient restart of reversed forks.

The manuscript addresses a very important question in the field, which is how RAD51-mediated fork reversal is catalyzed and which proteins are involved in the regulation of this process. The manuscript is very well written and the limitations of the study are discussed appropriately.

The strengths of the study are that isogenic human cell lines were used. Moreover, further validation experiments were conducted in an untransformed human cell line. Observed phenotypes were challenged by tests for antagonistic effects when compared to known factors involved in replication fork remodeling (e.g., ZRANB3 knockdown).

We are grateful to this reviewer for appreciating the relevance of the biological question addressed and importance of our observations.

I have three minor comments:

1. Reference #53 seems inappropriately cited as this reference deals with the effects of PARP-1 on forks (not with a paralog).

In this paper, besides the role of PARP-1 on forks, Sugimura et al. used XRCC3-defective DT40 cells to test the impact of an HR defect on fork progression, reporting an altered fork progression rate upon CPT treatment (Figures 5 and S3). While trying to summarize the scattered information currently available in vertebrates on the role of RAD51 paralogs upon replication stress, it thus seemed a relevant paper to cite.

2. Somyajit K et al. NAR (2015), already showed that XRCC2 is dispensable for fork restart and that X3/C promote fork restart. The investigators cite this paper and state in this context – "...that the functional role of these factors in replication has remained elusive." I suggest that this statement be omitted or toned down (also given the following sentence which again cites ref #51).

We have now toned down the sentence mentioned by the reviewer, and changed it into "...the mechanistic role(s) of these factors in replication were not investigated systematically".

3. RAD51 downregulation would be expected to negatively impact fork restart, but does not do so after 2 mM HU for 2h. Under these conditions, forks restart faster when RAD51 is depleted. Is it expected that the restart of these stalled forks does not require any RAD51, or is the remaining RAD51 sufficient to do so?

We are aware these results might be at odd with other reported data (e.g. Petermann E. et al. Mol Cell 2010), showing fork restart problems upon RAD51 defects. It should be noted, however, that RAD51 depletion rapidly impairs cell growth, finally resulting in cell cycle arrest and cell lethality upon prolonged downregulation. In order to overcome this limitation and possibly rule out any indirect cellular effect of sustained RAD51 depletion on fork restart proficiency, we have shortened the downregulation time to 24 hours, which allows satisfactory RAD51 depletion and induces marked defects in fork progression and remodelling, without yet impairing cellular fitness (Zellweger et al. JCB 2015). We do not observe a requirement of RAD51 for efficient fork restart in these experimental conditions, while the published data reporting fork restart defects upon RAD51 depletion were obtained upon prolonged downregulation (>48 hr). We have now specified this technical point, while describing the data in Fig. 4a (page 12, bottom). That said, we cannot formally rule out that residual RAD51 protein levels upon short downregulation may be sufficient to sustain efficient fork restart. Indeed, different extents of RAD51 downregulation have already been reported to drive opposite outcomes in terms of fork protection (Bhat KP. et al. Cell Report (2018), reflecting mild vs strong impairment of fork remodelling. It is thus technically possible that similar effects apply to RAD51 function in fork restart.

Reviewer #2 (Remarks to the Author):

In the manuscript by Berti M. and Teloni F. et al., the authors identified the human RAD51 paralogs (RAD51B, RAD51C, RAD51D, XRCC2 and XRCC3) from a targeted siRNA library in U2OS and RPE-1 cells under replication stress. While previous work has investigated the role of the RAD51 paralogs in canonical DSB repair, little cellular investigation has been performed in other settings of genome stability. The authors showed that the BCDX2 components RAD51C and RAD51D enable fork slowing and their depletion results in a lower percentage of reversed forks. Next, the authors created CRISPR KO cells of RAD51C, RAD51D, and XRCC3 and demonstrate in BRCA2 depleted cells that the BCDX2 complex components RAD51C and RAD51D are essential for restoring fork stability. The authors find that XRCC3 depletion increased the percentage of stalled forks and that the CX3 complex is important for replication restart. In conclusion, the authors provide evidence that the RAD51 paralog complexes have specific and distinct functions under conditions of replication fork stress (CPT vs HU). This important insight suggests that the two subcomplexes work sequentially to overcome replication stress. Their model suggests that the RAD51 paralog complex BCDX2 restrains replication progression while the CX3 complex mediates replication restart. These new exciting studies provide insight regarding RAD51 regulation.

We thank this reviewer for the appreciation of our work and its relevance.

Major comments:

1) Given the model in which BCDX2 acts early and allows downstream action of CX3 and BRCA2, does loss or depletion of RAD51B, RAD51D, or XRCC2 result in reduced BRCA2 or XRCC3 at the fork? Perhaps iPOND or foci formation upon replication stress would more directly demonstrate this model.

We agree with this reviewer that this would be an important prediction of our model. Monitoring the recruitment of these factors to forks, one would expect the recruitment of downstream factors (BRCA2, XRCC3) to depend on upstream ones (e.g. BCDX2). To our knowledge, no good tool (antibody, staining condition) is currently available to reliably monitor by IF staining BRCA2 recruitment to forks. Furthermore, the protein is far too big and unstable for proper detection under harsh iPOND experimental conditions (extensive formaldehyde crosslinking and its reversal by prolonged boiling in WB loading buffer). iPOND-MS studies have also failed to reliably detect BRCA2 enrichment at fork, as recently shown in a comprehensive survey of protein enrichment at replication forks (Wessel SR. *et al.* Cell Report 2019). Besides technical difficulties, this may reflect the role of BRCA2 as accessory factor in RAD51 loading, which – despite its crucial function in fork integrity - may not imply stable, detectable association of BRCA2 with replication factories. Finally, since iPOND-MS results are expressed as ratio between short EdU labelling and thymidine chase samples, a fairly abundant BRCA2 accumulation at postreplicative chromatin (for postreplicative gap protection for example) might mask its active fork recruitment.

Regarding fork recruitment of RAD51 paralogs (and in particular of XRCC3) – a crucial prerequisite to address the reviewer's question – we have tried very hard to achieve this goal, both before submission, and also extensively during revision, but faced significant technical difficulties. We are aware, from ongoing discussions with colleagues, that this is a shared problem with numerous labs, likely reflecting the low abundance of RAD51 paralogs, their possibly transient association with replicating chromatin and/or the lack of highly specific tools for their detection.

All our attempts to monitor RAD51 paralog recruitment at forks are summarized here below.

1. Foci detection by IF. We have tested a number of commercially available antibodies and various staining conditions, but were consistently unable to detect specific nuclear foci, which would disappear in the KO cell lines (data not shown). Please note that some of the antibodies previously used to reveal nuclear foci of RAD51 paralogs are no longer produced (Santacruz, see for example Somyajit et al., NAR 2015). Furthermore, even assuming successful detection of nuclear foci for these proteins, in our view this approach would not confidently allow monitoring fork recruitment, as opposed to possibly marking postreplicative or even replication-independent HR events, where these proteins have already been shown to act.
2. Proximity ligation assay (PLA) with nascent DNA or replisome components. We reckoned that proximity-ligation assays – where signals rely on an amplification step – may allow detecting even minimal levels of RAD51 paralogs, when recruited in close proximity to replisome components or nascent DNA, as successfully achieved for RAD51 itself and its association with PCNA or incorporated EdU (Fig. R2a and R2b). Nonetheless, multiple attempts to reveal specific recruitment of RAD51C and/or XRCC3 (by using both specific paralog antibodies or FLAG antibody in reconstituted KO U2OS cells) to PCNA/EdU were unsuccessful, yielding merely unspecific signals (Fig. R2a and R2b).

3. iPOND. We have also tried to use iPOND in HEK293T cells to reveal the fraction of RAD51 paralogs bound to replication forks, comparing nascent DNA and mature chromatin, and including optional HU arrest and release (Fig. R3a). All controls (PCNA, H3, γ H2AX) behaved as expected, and a minor fraction of RAD51C and RAD51D could be detected at forks, with only marginal variations induced by the HU treatment (Fig. R3b). However, XRCC3 detection – which was crucial to possibly monitor the sequential recruitment of the two complexes – proved impossible with the currently available antibodies. Importantly, XRCC3 was also undetectable by iPOND using XRCC3-KO HEK293T cells complemented with FLAG-XRCC3, despite higher expression than the endogenous level (see Fig. 3), and even when using the FLAG antibody (Fig. R3c). It is likely that this failure in XRCC3 detection reflects particularly poor commercially available antibodies and difficulties in using the FLAG antibody for iPOND western blot (possibly due to the short peptide epitope modified during crosslinking). Even in standard WBs on cell lysates, XRCC3 detection requires far longer exposures than for the other paralogs. Moreover, these antibodies, mostly monoclonal, are possibly unable to recognize the antigen upon crosslinking condition of iPOND. Besides these technical issues, it is also possible that - differently from BCDX2, which may be low-abundant, but a stable component of the replisome - XRCC3 recruitment could be particularly transient, as it may be specifically recruited to drive fork restart, leaving forks as soon as this is achieved. Importantly, a recent systematic iPOND-MS study, coupled to an extended bioinformatics analysis of previously published iPOND data, identified almost 600 proteins stably and consistently bound to replication forks, and none of the RAD51 paralogs was found among these proteins (Wessel SR. *et al.* Cell Report 2019).
4. Taking advantage of an ongoing collaboration with Prof. Eli Rothenberg (NYU, USA), we have also tried to monitor recruitment of RAD51 paralogs to forks using super-resolution microscopy (STORM). The Rothenberg group has been able to show recruitment of various DNA repair factors at stalled and broken forks, including rapid recruitment of RAD51 and RAD52 to forks in response to mild replication stress (Fig. R4a; Whelan *et al.*, Nature Comms 2018; Whelan *et al.*, manuscript submitted). Unfortunately, when the Rothenberg lab applied the same methodology for the detection of RAD51 paralogs – using currently available antibodies – failed to reveal any significant colocalization of these proteins with nascent DNA (EdU) as compared to random distribution, both in presence and absence of HU (Fig. R4b). Very similar unexciting negative results were obtained by: a) attempting colocalization of RAD51 paralogs with MCM proteins or PCNA, b) varying the duration of the HU treatment, c) using CPT as alternative drug for replication interference (data not shown). Also by this method, it seems most likely that currently available antibodies are not capable to reveal low levels of these accessory factors specifically recruited to forks. Overcoming these technical limitations will most likely require tagging of endogenous RAD51 paralogs, using tags that recently proved particularly potent for live cell imaging via super-resolution microscopy. However, obtaining and optimizing these cellular tools for these imaging experiments will require significant additional time and effort in the Rothenberg lab, and is clearly beyond the scope of this manuscript.

In light of the present limitations to monitor the timing of recruitment of these factors, we have thus capitalised on the power of genetic experiments, in order to further test (and support) the sequential role of these proteins at replication forks. Besides several lines of evidence already included in our first submission, we obtained and included in revision one more important clear-cut result (see point 3 here below) of the upstream role of BCDX2 vs CX3, which is now supported in the revised manuscript by multiple independent experiments:

1. BCDX2 inactivation (Figs. 2, 3, S2 and S3) phenocopies RAD51 downregulation (Zellweger *et al.*, JCB 2015; Mijic *et al.*, Nature Comms 2017), in terms of fork slowing, fork protection and fork reversal, clearly implying this subcomplex in the early steps of fork remodelling.
2. As for RAD51 and ZRANB3 (Mijic *et al.*, Nature Comms 2017), impairment of BCDX2 (by RAD51C or RAD51D-inactivation) suppresses fork degradation in BRCA2 cells (Figs 3 and S3).
3. As for RAD51 and ZRANB3 (Figs. 4b-c and S4d-e in the submitted ms), BCDX2 impairment by RAD51D downregulation (Figs. 4d and S4f in the revised ms) suppresses fork degradation in BRCA2 cells, placing this subcomplex clearly upstream of protection/restart. Importantly, RAD51C downregulation per se – which drastically reduces the levels of both RAD51C and XRCC3 – shows no restart defect, further confirming that the fork restart defect linked to XRCC3 deficiency is suppressed by preventing fork reversal via the RAD51C defect.

We believe this evidence overall strongly supports the key claim of this manuscript, regarding the sequential role of these two complexes in fork remodelling and restart. Nonetheless, we now acknowledge in the revised manuscript that – despite all the attempts described above – we were unable to monitor directly their recruitment to replication forks (page 17). In light of this, we avoided any statement in the manuscript explicitly referring to their physical recruitment, rather emphasizing their sequential function at stalled replication forks. We have also avoided any distinction between “early vs late” roles, preferring “upstream vs downstream” functions.

2) In the model and introduction, the authors should include the recent description of RAD52 and SMARCAL1 in fork protection (nascent strand) and restart (Malacaria et al., Nature Communications 2019). This is another important example of an HR protein playing noncanonical HR functions. Moreover, the function of RAD52-SMARCAL1 becomes increasingly more important upon BRCA2 deficiency and should be included in the model.

Indeed, this manuscript extends the notion of a non-canonical role of HR factors at stalled forks and we are now referring to it in the introduction. However, we have centred our working model around the novel data presented here. Graphical representation of all key factors involved in fork remodelling would in fact require addition of many more factors than the two cited by the reviewer and would end up diluting the message, diminishing the graphical impact of the model.

3) While under consideration for publication, characterization of the human RAD51 paralog containing complex, the Shu complex, was published (Martino J. et al., Nucleic Acids Research 2019) and is referenced several times throughout the text. Please modify the sentence, "Dedicated studies will be required to explore a possible role for the Shu complex components SWSAP1 and SWS1 in the replication stress response" to include their published role in replication restart (Martino J. et al., Nucleic Acids Research 2019).

We have now modified the cited sentence (page 8, bottom), to include a reference to this important recent publication.

4) Explicitly state the cell line used for the siRNA screening in your description of Figure 1.

The cell line (U2OS) used for the siRNA screen is now explicitly mentioned in the main text (page 7) and in the legend of Figure 1.

5) It is confusing that the authors state that DSBs are not formed under 50 nM CPT treatment but then state "Impairing fork remodeling activities may not only negatively affect RAD51 binding to forks, but can also induce chromosomal breakage and thereby enhance RAD51 accumulation in foci." Please address for consistency.

What we meant to propose here is that genetic impairment of fork remodeling may induce a certain level of fork breakage (and associated RAD51 loading) even at CPT doses that do not detectably induce fork breakage in control cells. To clarify this concept, we have replaced the original sentence with the following one (page 8): "Impairing fork remodeling activities may not only negatively affect RAD51 binding to forks, but can also induce fork breakage even at these minimal CPT doses, thereby enhancing RAD51 accumulation in foci".

6) sgSWSAP1 and sgSWS1 cells have been assessed for RAD51 foci formation previously (Martino J. et al., Nucleic Acids Research 2019).

As mentioned above (point 3), the reference to this recent paper was added while discussing the role of the Shu complex in RAD51 loading at stalled forks.

7) In Figure 1c, SHFM1 is better known in the literature as DSS1 and has been published as this protein name in association with BRCA2, RPA etc.

We thank the reviewer for this suggestion. We have now replaced SHFM1 by DSS1 in Fig. 1c and Tables S1 and S2.

8) In Figure 1h, XRCC3 forms more RAD51 foci upon CPT and IR treatment relative to the other paralog knockdowns. This does not appear to be specific to its role in replication stress but could be due to a less efficient knockdown or a lower overall protein abundance needed for its cellular function. The siXRCC3 data actually mirrors the siControl. The authors should show this in their CRISPR knockouts as it is one of the hits they interrogate.

We thank this reviewer (and reviewer 3) for noticing this discrepancy, which led us to revisit this point and have a closer look at the effect of XRCC3 knockdown on RAD51 foci formation. These data are generated through IF stainings, which in terms of absolute signal intensities vary from one experiment to the next, making relatively subtle effects difficult to evaluate for statistical significance. We have now calculated a fold change for siXRCC3 in respect to control siRNA from three independent experiments, which yielded an

average reduction to 69% of control for one oligo and an average reduction to 83% for a second oligo (Fig. R1a, for reviewers only). Although the variation in signal intensity allows to reach statistical significance only for one of the two oligonucleotides, these data consolidate a trend for a role of XRCC3 in RAD51 loading, albeit minor compared to the other RAD51 paralogs. We have now selected for Fig. 1f a similar panel from a different experiment, where the difference between control siRNA (siCtrl) and siXRCC3 is more evident. We have also included an additional analysis in Fig. R1b, which is focused on a comparison of S-phase cells only (red data points in Fig. 1f), making the difference between siCtrl and siXRCC3 more evident. Importantly, the effect of XRCC3 Knock-Out (KO) by CRISPR - and that of its genetic reconstitution - on RAD51 loading were already shown in the submitted manuscript (and are now displayed in Fig. S2d): the contribution of XRCC3 to RAD51 loading in this context is indeed more important, in agreement with Garcin et al., 2019. However, this is clearly the case also for the other tested RAD51 paralogs (RAD51C and RAD51D). Rather than residual XRCC3 levels upon siRNA-mediated knockdown (which seem marginal, see Fig. 2b), the stronger effects of RAD51 paralog KO vs siRNA-mediated downregulation is likely to reflect chronic vs acute gene inactivation, which may reveal long-term effects of the inactivation of these factors.

Overall, we are convinced that our data – refined in revision based on the reviewers' comments – allow us to conclude that XRCC3 inactivation has a detectable effect on RAD51 loading, albeit minor compared to other RAD51 paralogs. This in turn sets the stage for the functional analyses in the following figures, which strongly support a different mechanistic role for CX3 vs BCDX2 in response to replication stress.

9) In Figure 2, while the study clearly assessed protein expression, the “stability” was not directly assessed. Either perform cycloheximide chases to demonstrate protein stability or remove “stability” from the description for Figure 2.

We have now avoided the term “stability” while describing Figure 2a, simply referring to the effects of different downregulations on the levels of other complex subunits.

10) In Figure 4b, if BCDX2 and CX3 are functioning sequentially, then the RAD51 and XRCC3 co-depletion should have the same phenotype as RAD51B, RAD51D, or XRCC2 co-depletion with XRCC3. This experiment would strengthen the evidence for distinctive roles for the subcomplexes.

We thank this reviewer for this inspiring suggestion. We have now performed the suggested experiment, by depleting RAD51D (the BCDX2 specific subunit that we inactivated in all other mechanistic experiments) and assessing whether this may suppress – as predicted by our model – the fork restart defect due to XRCC3 downregulation. The result of this experiment is essentially a phenocopy of what we already observed for RAD51- or ZRANB3-depletions - a complete rescue of fork restart efficiency and velocity in XRCC3-downregulated cells. These additional data significantly support to the sequential role of BCDX2 and CX3 in fork remodelling and restart (i.e. the key take-home message of this manuscript), and are hence now included in Fig. 4d and Fig. S4f and described in page 13.

Minor comment:

1) “Besides the possibility of incomplete downregulation, this result may also reflect ...” Is it possible that these factors may not have been identified in your screen as they play a role in persistent fork stalling and/or collapse?

Yes, indeed. As we have tried to explain in the text, it is possible that several candidates in our screen did not score as possible expected (in respect to their functional role on RAD51 loading), because of their role at stalled or collapsed forks. In light of such role, the molecular consequences of their downregulation in terms of fork integrity may indirectly affect the extent of RAD51 chromatin loading, via DSB formation.

Reviewer #3 (Remarks to the Author):

The manuscript "Sequential role of RAD51 paralog complexes in replication fork remodeling and restart" By Berti et al. describes the roles of the RAD51 paralogs homologues recombination factors (RAD51B, RAD51C, RAD51D, XRCC2 and XRCC3) in DNA replication. These paralogues form two distinct complexes the BCDX2, containing RAD51B-D and XRCC2, and CX3 complex containing RAD51C and XRCC3. The authors nicely show that BCDX2 functions to slow down fork progression and promote fork reversal while CX3 complex is not essential for fork reversal but is important for fork restart.

The experiments and the data presented in the manuscript were done in a thorough manner and the data is very interesting.

We are grateful to Dr. Aharoni for the positive evaluation of our work.

In order to clarify the manuscript more, I suggest the following corrections:

1. I found the introduction much too long, while it is important to provide a thorough background and describe previous works, the authors should focus on the most relevant previous studies and open questions in the field, rather than providing a too long and elaborated account of the background. I urge the authors to at least cut ~30% of the introduction to allow for a more efficient and fast reading of the paper. For example, the last sentence of the first paragraph in page 4: "In addition to..." in my view is not important.

That specific sentence has been deleted, and the Introduction has now been reduced by approximately 150 words (ca. 15%). While reporting new mechanistic data on a class of proteins that has been studied for over two decades, we consider essential not only to mention previously published observations, but also to discuss their possible limitations and to explain the mechanistic relevance of our investigations in an isogenic genetic background. We thus found it impossible to shorten Introduction any further, without weakening completeness and clarity of the manuscript.

2. Results section page 7: it is not clear what is the size of the siRNA library, how many proteins were targeted in the screen?

The size of the library – which is directly visible in Supplementary Table 1 - is now also specified in the main text (page 7).

3. Page 7 bottom: Define what is a z-score for the general reader?

We have now introduced the following statement in the main text: "Reassuringly, when ranked according to their deviation from the mean (z-score), ...".

4. Page 10 bottom, "Importantly, defective fork slowing..." This is a poor phrasing, please rephrase, faster fork progression as an example.

We have now replaced "defective fork slowing" with "unrestrained fork progression".

5. Discussion section, page 15: second paragraph, it's good to remind the reader the nature of the two complexes, BCDX2 composed of and CX3 composed of...., since all the data presented target subunits of these complexes (RAD51C, RAD51D and XRCC3).

We thank the reviewer for this important suggestion. We have now recalled the composition of the two complexes in the first paragraph of the Discussion, and summarized the tools used to selectively inactivate one and/or the other complex (page 15).

6. Page 17 first section. Describing the sequential role of the complexes. It is not clear what are the experimental evidences supporting this sequential role and whether the authors can prove which complex is recruited first and which one later on.

We kindly refer to our response to reviewer 2, point 1, where we describe our numerous attempts to directly monitor the recruitment kinetics of these factors to replication forks. Although we were unable to demonstrate sequential recruitment, we now have several lines of strong evidence for the sequential function of BCDX2 and CX3 in replication fork remodelling and restart:

- 1) BCDX2 inactivation (Figs. 2, 3, S2 and S3) phenocopies RAD51 downregulation (Zellweger et al., JCB 2015; Mijic et al., Nature Comms 2017), in terms of fork slowing, fork protection and fork reversal, clearly implying this subcomplex in the early steps of fork remodelling.
- 2) As for RAD51 and ZRANB3 (Mijic et al., Nature Comms 2017), impairment of BCDX2 (by RAD51C or RAD51D-inactivation) suppresses fork degradation in BRCA2 cells (Figs 3 and S3).
- 3) As for RAD51 and ZRANB3 (Figs. 4b-c and S4d-e in the submitted ms), BCDX2 impairment by RAD51D downregulation (Figs. 4d and S4f in the revised ms) suppresses fork degradation in BRCA2 cells, placing this subcomplex clearly upstream of protection/restart. Importantly, RAD51C downregulation *per se* – which drastically reduces the levels of both RAD51C and XRCC3 – shows no restart defect, further confirming that the fork restart defect linked to XRCC3 deficiency is suppressed by preventing fork reversal via the RAD51C defect.

We believe this evidence overall strongly supports the key claim of this manuscript regarding the sequential role of these two complexes in fork remodelling and restart. Nonetheless, we now acknowledge in the revised manuscript that – despite all the attempts described above– we were unable to monitor directly their recruitment to replication forks (page 17). In light of this, we avoided any statement in the manuscript explicitly referring to their physical recruitment, rather emphasizing their sequential function at stalled replication forks. We have also avoided any distinction between “early vs late” roles, preferring “upstream vs downstream” functions.

7. Page 17 fork restart, can the authors provide more mechanistic evidences regarding how fork restart is promoted, with respect to replisome recruitment and DNA synthesis?

We agree that, even within our manuscript, the detailed mechanism for CX3-mediated fork restart remains elusive, yet would be an exciting research venue to explore. As stated in the main text, we are convinced that biochemical reconstitution of the restart reaction, including several known fork restart activities, will be crucial to decipher CX3-mediated restart mechanism in all its detail. We are aware that several research groups are currently attempting this ambitious task, but this is certainly a project on its own (and a very complex one!), which lies beyond the focus of this manuscript. We have now added a sentence to the discussion including an additional hypothesis on how CX3 could assist restart of reversed forks (page 18), emphasizing the need for further biochemical investigations to clarify this aspect.

8. Fig. 1, it is not clear how XRCC3 was identified? Fig. 1f show the levels of RAD51 foci of si XRCC3 relative to si control and the level of foci looks very similar. While the authors refer to the milder effects of XRCC3 downregulation in page 9, it is important to show statistical tests between the samples in 1f to test significance differences between different siRNA samples.

We thank this reviewer (and reviewer 2) for noticing this discrepancy, which led us to revisit this point and have a closer look at the effect of XRCC3 knockdown on RAD51 foci formation. These data are generated through IF stainings, which in terms of absolute signal intensities vary from one experiment to the next, making relatively subtle effects difficult to evaluate for statistical significance. We have now calculated a fold change for siXRCC3 in respect to control siRNA from three independent experiments, which yielded an average reduction to 69% of control for one oligo and an average reduction to 83% for a second oligo (Fig. R1a, for reviewers only). Although the variation in signal intensity allows to reach statistical significance only for one of the two oligonucleotides, these data consolidate a trend for a role of XRCC3 in RAD51 loading, albeit minor compared to the other RAD51 paralogs. We have now selected for Fig. 1f a similar panel from a different experiment, where the difference between control siRNA (siCtrl) and siXRCC3 is more evident. We have also included an additional analysis in Fig. R1b, which is focused on a comparison of S-phase cells only (red data points in Fig. 1f), making the difference between siCtrl and siXRCC3 more evident. Importantly, the effect of XRCC3 Knock-Out (KO) by CRISPR - and that of its genetic reconstitution – on RAD51 loading were already shown in the submitted manuscript (and are now displayed in Fig. S2d): the contribution of XRCC3 to RAD51 loading in this context is indeed more important, in agreement with Garcin et al., 2019. However, this is clearly the case also for the other tested RAD51 paralogs (RAD51C and RAD51D). Rather than residual XRCC3 levels upon siRNA-mediated knockdown (which seem marginal, see Fig. 2b), the stronger effects of RAD51 paralog KO vs siRNA-mediated downregulation is likely to reflect chronic vs acute gene inactivation, which may reveal long-term effects of the inactivation of these factors.

Overall, we are convinced that our data – refined in revision based on the reviewers’ comments – allow us to conclude that XRCC3 inactivation has a detectable effect on RAD51 loading, albeit minor compared to other RAD51 paralogs. This in turn sets the stage for the functional analyses in the following figures, which strongly support a different mechanistic role for CX3 vs BCDX2 in the response to replication stress.

9. Fig. 4a, it is not clear what are the examples of fork restart and fork stalling, from what cell line these examples

were taken. In addition, what is the criteria for fork stalling? Specifically, what ratio of green/red signal determine the threshold for counting it as fork stalling.

The genetic conditions from which the DNA fiber examples have been taken are now clearly indicated in the figure. The criterion used to define fork stalling has been arbitrarily set to a ratio (length of green vs red) of < 0.1. This has now been specified in the Material and Methods section.

10. Figure 5b, the model, I think that a better choice of colors can allow the reader to more clearly identify the two complexes.

We thank this reviewer for this useful suggestion, which – we agree – will help the readers to grasp the key take-home message from the graphics. Both in Fig. 2 and in Fig. 5, we now opted for blue tints for BCDX2, and violet tints for CX3, with a light violet for the common subunit (RAD51C), keeping BRCA2 clearly distinguishable (green).

Amir Aharoni

a

	siCon	siXRCC3 #1	fold change	siXRCC3	fold change
Experiment 1	14.68	11.17	0.76	11.36	0.77
Experiment 2	14.88	12.39	0.83	12.97	0.87
Experiment 3	7.8	3.76	0.48	6.54	0.84

Average	0.69	Average	0.83
Stand. Dev.	0.19	Stand. Dev.	0.05
p-value	0.45	p-value	0.04

b
Figure R1 (for reviewers only). Role of XRCC3 in RAD51 loading upon mild CPT treatment. (a) Fold change reduction of RAD51 foci in siXRCC3- versus siControl (siCtrl)-downregulated U2OS cells, treated with CPT (50 nM for 45 min). The table shows average and statistical parameters of RAD51 foci formation from three independent QIBC experiments, with two different siXRCC3 oligos. **(b)** Influence of RAD51 paralog downregulation on RAD51 foci formation in S-phase U2OS cell, treated with CPT (50 nM for 45 min; red data points in Fig. 1f).

Figure R2 (for reviewers only). Testing the association of RAD51 paralogs with replication forks by Proximity ligation assay (PLA). (a) U2OS wildtype (WT) or RAD51C-, RAD51D- and XRCC3-KO cells with reconstitution of the respective FLAG-tagged protein (+), were untreated or treated with HU (2 mM for 2 hr) and tested for FLAG(rabbit)-PCNA(mouse) PLA foci formation by QIBC. The scatter plots show the cell-cycle distribution (based on DNA content) of PLA foci in the different genetic conditions. No S-phase specific increase in PLA signals is detected between PCNA and any of the tested RAD51 paralogs. RAD51(rabbit)-PCNA(mouse) PLA in WT cells is used as technical control under the same experimental conditions, providing evidence that the approach is capable of revealing known PCNA interactors. (b-d) Scatter plots showing the cell-cycle distribution (based on DNA content) of PLA foci, detecting the colocalization of target proteins with nascent DNA (EdU). (b) RAD51(rabbit)-EdU(mouse) PLA in U2OS cells untreated or treated with HU (2 mM for 2 hr); this technical control shows HU-induced RAD51 recruitment to forks, providing evidence that the technique is capable of revealing specific association with nascent DNA. (c) RAD51C (mouse)-EdU(rabbit) PLA in U2OS cells untreated or treated with HU (2 mM for 2 hr). Low levels of PLA foci detected in S phase cells are not increased upon HU-induced fork stalling. (d) RAD51C(FLAG rabbit)-EdU(mouse) PLA in U2OS wildtype (WT) or RAD51C-KO cells with reconstitution of the FLAG-tagged protein (+). Low levels of PLA foci detected in S phase cells are visible also in the cell line that does not bear FLAG-RAD51C, and are most likely unspecific signals associated with EdU incorporation *per se*.

Figure R3 (for reviewers only). Testing the association of RAD51 paralogs with replication forks by immunoprecipitation of protein on nascent DNA (iPOND). (a) Schematic of the different EdU-labelling and treatment conditions used for iPOND experiments showed in Fig. R3b and R3c. (b) and (c) Normal HEK293T cells or XRCC3-KO HEK293 cells stably reconstituted with FLAG-XRCC3 were differentially EdU-labelled and treated with HU. Proteins associated with replication forks were isolated by standard (not native) iPOND procedure and detected with the indicated antibodies. The thymidine (Thy) chase experiment is used to discriminate proteins associated with chromatin behind replicating forks. In the control experiment, cells are treated with DMSO instead of EdU (No EdU). As expected PCNA is released from forks upon HU treatments, which induce γ H2AX at forks. RAD51C and RAD51D show some fork association, regardless of the treatment. XRCC3 is not detectable at replication forks, either by XRCC3 antibodies (b and c) or by FLAG antibodies in a cell line expressing relatively high levels of FLAG-XRCC3 (c; see also Fig. 2D)

[REDACTED]

REVIEWERS' COMMENTS:

Reviewer #2 (Remarks to the Author):

The authors have now adequately addressed our concerns with a minor edit summarized below. In response to our Major Comment 8, we recommend that the authors include Figure R1B (not A) in the publication. The S phase data suggests these RAD51 foci are actually repairing replication fork-associated damage, which is not impacted by XRCC3. This should be added to Supplementary Figure 2 because I found this data more compelling than Figure 1g/h.

Reviewer #3 (Remarks to the Author):

The points that I raised in my review have been addressed well. I have no additional comments for this nice manuscript.

REVIEWERS' COMMENTS:

Reviewer #2 (Remarks to the Author):

The authors have now adequately addressed our concerns with a minor edit summarized below. In response to our Major Comment 8, we recommend that the authors include Figure R1B (not A) in the publication. The S phase data suggests these RAD51 foci are actually repairing replication fork-associated damage, which is not impacted by XRCC3. This should be added to Supplementary Figure 2 because I found this data more compelling than Figure 1g/h.

Response: We thank this reviewer for the support. We have now added Figure R1B from our rebuttal as final panel (e) of Supplementary Figure 1, where all other validation data of our QIBC screen are included.

Reviewer #3 (Remarks to the Author):

The points that I raised in my review have been addressed well. I have no additional comments for this nice manuscript.

Response: We thank this reviewer for the support.